# Early detection of emerging viral variants through analysis of community structure of coordinated substitution networks

Fatemeh Mohebbi [1,2], Alex Zelikovsky[1], Serghei Mangul [2,3], Gerardo Chowell[4] & Pavel Skums [1,5] ✉

The emergence of viral variants with altered phenotypes is a public health challenge underscoring the need for advanced evolutionary forecasting methods. Given extensive epistatic interactions within viral genomes and known viral evolutionary history, efficient genomic surveillance necessitates early detection of emerging viral haplotypes rather than commonly targeted single mutations. Haplotype inference, however, is a significantly more challenging problem precluding the use of traditional approaches. Here, using SARS-CoV-2 evolutionary dynamics as a case study, we show that emerging haplotypes with altered transmissibility can be linked to dense communities in coordinated substitution networks, which become discernible significantly earlier than the haplotypes become prevalent. From these insights, we develop a computational framework for inference of viral variants and validate it by successful early detection of known SARS-CoV-2 strains. Our methodology offers greater scalability than phylogenetic lineage tracing and can be applied to any rapidly evolving pathogen with adequate genomic surveillance data.

Understanding the predictability of evolution and the relative impact of random and deterministic factors in evolutionary processes is a fundamental problem in life sciences. This problem gains an applied significance in the context of viruses and other pathogens, as even a modest degree of predictability of pathogen evolution can enhance our ability to forecast and, therein, control the spread of infectious diseases[1–4].

The most evident example of the importance of this problem is the case of severe acute respiratory syndrome coronavirus 2 (SARS-CoV-2). The successive waves of COVID-19 are driven by the emerging genomic variants of interest (VOIs) or variants of concern (VOCs) that have been associated with altered phenotypic features, including transmissibility[5,6], antibody resistance, and immune escape[7–9]. Each genomic variant is defined as a phylogenetic lineage characterized by a specific combination of single amino acid variants (SAVs) and/or indels

acquired over the course of SARS-CoV-2 evolution. For instance, lineages B.1.1.7 (alpha variant by WHO classification) and B.1.617.2 (delta variant) are defined by distinct families of 7 SAVs in the S gene decoding the spike protein, many of which have been linked to enhanced fitness compared to preceding SARS-CoV-2 lineages[6,10–13].

Genomic epidemiology has been crucial for monitoring the emergence and spread of SARS-CoV-2 variants since the start of the COVID-19 pandemic. SARS-CoV-2 genomes sampled around the globe and produced using high-throughput sequencing technologies have been analyzed by a plethora of phylogenetic, phylodynamic, and epidemiological models[14] to detect spreading lineages and measure their reproductive numbers and other epidemiological characteristics. However, these methods, powerful and valuable as they are, are primarily applied retrospectively. In other words, they allow to detect growing lineages and measure their fitness only when these lineages

[1]Department of Computer Science, Georgia State University, Atlanta, GA, USA. [2]Titus Family Department of Clinical Pharmacy, USC Alfred E. Mann School of Pharmacy and Pharmaceutical Sciences, University of Southern California, Los Angeles, CA, USA. [3]Department of Quantitative and Computational Biology, USC Dornsife College of Letters, Arts and Sciences, University of Southern California, Los Angeles, CA, USA. [4]School of Public Health, Georgia State University, Atlanta, GA, USA. [5]School of Computing, College of Engineering, University of Connecticut, Storrs, CT, USA. ✉e-mail: pavel.skums@uconn.edu

are already sufficiently prevalent. Moreover, existing phylogenetic and phylodynamic approaches are computationally expensive. They must use subsampling, simplifying assumptions, and heuristic algorithms without performance guarantees to handle the vast amounts of available genomic data (e.g., more than 14 million sequences in the GISAID database[15] at the time of submission of this paper). These considerations can impact their power, accuracy, and reliability.

In contrast to retroactive detection, the task of early detection or forecasting involves the proactive identification of SARS-CoV-2 genomic variants that have the potential to become prevalent in the future. This problem is more challenging as it is intertwined with the fundamental question of whether viral evolution can be predicted or whether one can replay the tape of life for the global SARS-CoV-2 evolution, using the metaphor of ref. [16]. For viruses, the possibility of evolutionary predictions remains a topic of debate[17]. Nevertheless, studies attempting to address the SARS-CoV-2 evolutionary forecasting problem have emerged[3,4,18–21]. Most of these studies have focused on the emergence of individual mutations, with some methods assuming that mutations accumulate independently or that the effects of their interactions can be averaged out over their genomic backgrounds[3,21].

Meanwhile, a number of studies have highlighted the significance of epistasis, i.e., the non-additive phenotypic effects of combinations of mutations, for SARS-CoV-2[4,22–27]. Using various methodologies, including phylogenetic analysis[23,26], direct coupling analysis[4], and in vitro binding measurements[24,27], these studies suggest the existence of an epistatic network that includes many genomic sites in the receptor-binding domain of the spike protein that is associated with increased binding affinity to angiotensin-converting enzyme 2 (ACE2) receptor[9,28,29]. Epistasis is closely linked to the complex structures of viral fitness landscapes[4,22,27,30], which determine the evolutionary trajectories of SARS-CoV-2 lineages and contribute to the high nonlinearity of its evolution, making forecasting challenging. The emergence of new variants of concern, such as the lineage B.1.1.529 (Omicron variant), is an example of such nonlinear phenomena[27]. Its rapid emergence does not align with the gradual mutation accumulation hypothesis and is still a topic of debate, with hypothesized origins including immune-suppressed hosts and reverse zoonosis[27,31–33].

Given the role of epistasis, it can be argued that selection often acts on combinations of mutations, or haplotypes, rather than on individual mutations. Therefore, effective forecasting should focus on viral haplotypes instead of solely on SAVs. However, predicting haplotypes is a significantly more challenging problem than predicting individual SAVs—in particular, simply due to the exponential increase in the number of possible haplotypes with genome length. This complexity precludes the use of traditional approaches utilized in most mutation-based studies, where a feature vector of epidemiological, evolutionary, and/or physicochemical parameters is calculated for each SAV, and a statistical or machine learning model is trained to predict SAV phenotypic effects. As a result, even studies that account for epistatic effects usually focus on assessing the phenotypic effects of individual mutations[4].

This paper focuses on predicting haplotypes of SARS-CoV-2 using a novel approach based on analyzing dense communities of the coordinated substitution networks of the spike protein, which reflects potential positive epistatic interactions[25,26,34]. We demonstrate that emerging haplotypes with altered phenotypes can be accurately predicted by leveraging these communities and introduce HELEN (Heralding Emerging Lineages in Epistatic Networks)—a variant reconstruction framework that integrates graph theory, statistical inference, and population genetics methods. HELEN was validated by accurately identifying known SARS-CoV-2 VOCs and VOIs up to months before they reached high prevalences and were designated by the WHO. Importantly, the majority of predictions were derived from data collected independently from different countries, further supporting their credibility. These results demonstrate that network density is a more precise, sensitive, and scalable measure than lineage frequency, allowing for reliable early detection or prediction of potential variants of concern before they become prevalent. Furthermore, the computational complexity of our method depends on genome length rather than the number of sequences, making it significantly faster than traditional phylogenetic methods for VOC detection and enabling it to handle millions of currently available SARS-CoV-2 genomes.

Our approach to the early detection of viral haplotypes utilizes a certain methodological similarity with the problem of inference of rare viral haplotypes from noisy sequencing data, particularly when produced by long-read sequencing technologies like Oxford Nanopore and PacBio. This problem has gained significant attention in recent years, with several new tools appearing each year[35–38]. Some of these tools accurately infer rare haplotypes with frequencies comparable to the sequencing noise level. In particular, several tools developed by the authors of this paper achieve such results by identifying and clustering statistically linked groups of SNV alleles[36,39,40]. Although this approach is not directly transferable to haplotype prediction, it provided a foundation for this study.

## Results
### Data
Genomic data and associated metadata analyzed in this study were obtained from GISAID[15]. This dataset includes 3,284,740 individual genome sequences available on GISAID up to April 26, 2022. We directly downloaded all available sequences and associated metadata from GISAID on April 26, 2022, accessible via https://doi.org/10.55876/gis8.230407vq. Our focus was on analyzing amino acid genomic variants of the SARS-CoV-2 spike protein, which is used for identifying variants of concern (VOC) Alpha (B.1.1.7), Beta (B.1.351), Gamma (B.1.1.28.1), Delta (B.1.617.2), Omicron (B.1.1.529.1) and variants of interest (VOI) Lambda (C.37), Mu (B.1.621), Theta (P.3), Eta (B.1.525), Kappa (B.1.617.1) by standard genomic surveillance tools adopted by WHO[41]. We extracted the spike protein alignment from the whole genome multiple sequence alignment, replacing ambiguous characters with gaps, and focused solely on SAVs while ignoring long indels. In order to better validate the predictive power of our approach, especially with respect to the Omicron lineage, we analyzed only sequences sampled before December 31, 2021, ~1 month after the designation of Omicron as a Variant of Concern by WHO. For defining VOCs and VOIs, we used the notations and lists of SAVs established by WHO[42]: a variant defined by SAVs at $k$ fixed genomic positions was associated with a $k$-haplotype with minor alleles (with respect to the standard Wuhan-Hu-1 (NC_045512.2) reference) at that positions. Variants epsilon (B.1.427), iota (B.1.526) and zeta (P.2), defined by 3–4 SAV, were excluded due to their short lengths.

Some analyzed sequences were labeled as under investigation by GISAID. In particular, these include some VOC/VOI sequences sampled earlier than the initial cases of these strains officially documented by WHO. In addition, we discovered some early-sampled sequences not marked as under investigation by GISAID. Consequently, our analysis was conducted on three distinct datasets:

(i) Complete set: Incorporates all sequences.
(ii) First truncated set: Excludes sequences labeled as under investigation.
(iii) Second truncated set: Excludes both sequences flagged as under investigation and early-sampled sequences that were not flagged.

The detection of linked pairs of SAVs and dense communities in coordinated substitution networks is affected by the number of sequences. Thus we focused on data from countries with the largest sample sizes, while maintaining geographic diversity. To do this, we selected two countries per continent (excluding Oceania) with the largest numbers of spike amino acid sequences sampled over the considered time period: the United Kingdom and Germany for Europe,

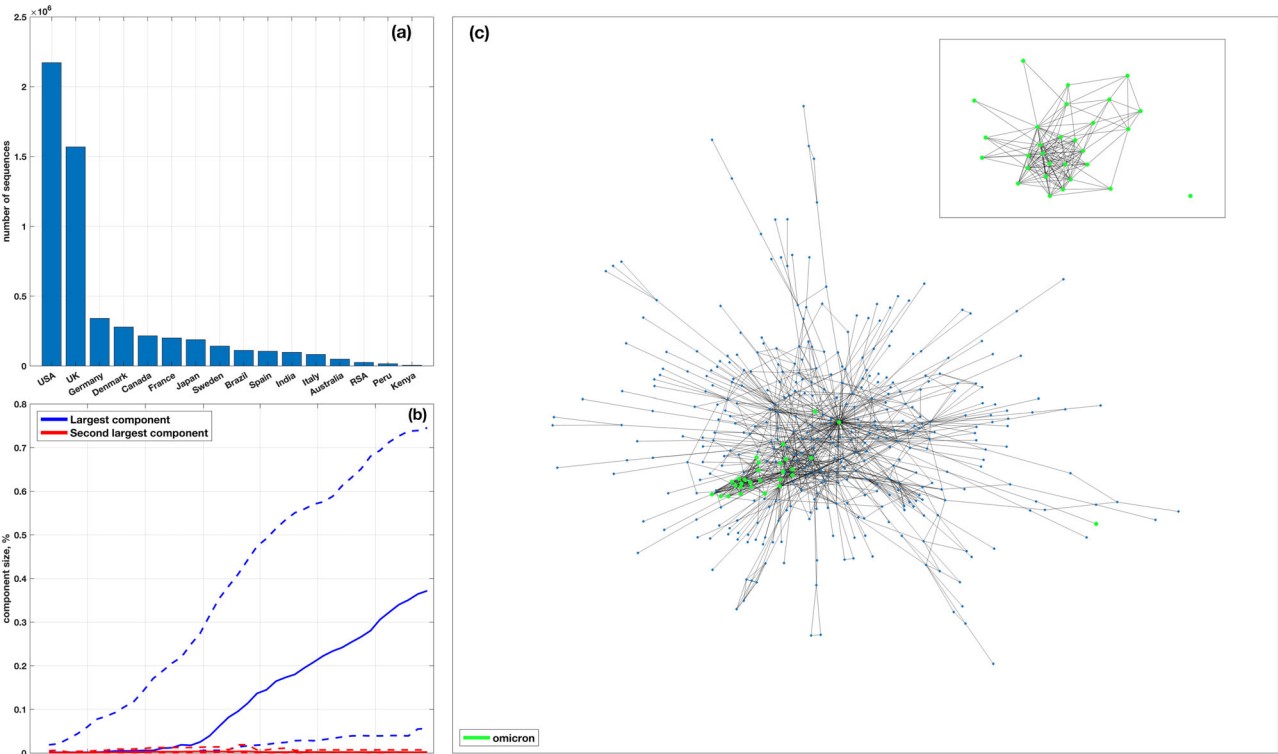

**Fig. 1 | Data and coordinated substitution networks. a** Numbers of analyzed spike amino acid sequences per country. **b** Relative sizes of the largest and second largest connected components of coordinated substitution networks over time. Solid and dashed lines depict median and maximum/minimum values over 16 countries at each time point, respectively. **c** An example of a giant component of a coordinated substitution network obtained using the complete dataset for the USA on January 11, 2021. The vertices highlighted in green correspond to SAVs of the Omicron variant (lineage B.1.1.529.1). Most of these SAVs form a dense community, visualizing the key idea of the study.

USA and Canada for North America, Brazil and Peru for South America, South Africa and Kenya for Africa, and Japan and India for Asia. In addition, we included Australia to represent Oceania and 5 extra countries with the largest samples, namely France, Denmark, Sweden, Spain, and Italy. Sequences from the selected countries were identified using GISAID metadata and analyzed separately. Thus, a total of 656 test cases (16 countries × 41 time points) have been considered. Figure 1a shows the analyzed sample sizes, which were not distributed uniformly, with the USA and United Kingdom accounting for approximately 66% of all sequences.

## The structure of S-gene-coordinated substitution networks

We utilized the method outlined in "Inference of coordinated substitution networks" to construct coordinated substitution networks for 16 countries at 41 uniformly distributed time points between May 1, 2020, and December 31, 2021 (with a 14-day difference between consecutive points). Initially, we evaluated the basic properties of these networks. We found that the majority of networks contained a single "giant" component (i.e., the connected component containing a significant fraction of graph vertices) that could include up to 75% of the vertices. Other connected components were significantly smaller ($P < 10^{-100}$, Kolmogorov–Smirnov test) and made up an average of 0.3% of the network size (Fig. 1b). Most of these smaller components consisted of isolated vertices.

Coordinated substitution networks of the S-gene tend to gradually evolve toward becoming scale-free, with a right-skewed power-law degree distribution. This type of network structure is often a result of a preferential attachment process, where a new vertex joining the network has a higher probability of connecting to an existing vertex with a higher degree. Indeed, to determine the best distribution fit for the observed degree distribution of the networks, we fitted negative binomial, beta negative binomial, Poisson, Yule–Simon, Generalized Pareto, and Pareto distributions, and compared their goodness of fit using the Bayesian Information Criteria. We found that the Yule–Simon, generalized Pareto and Pareto distributions, all describing a power-law, provided the best fit for ~55%, ~20%, and ~12% of networks, respectively. In addition, in all countries, the Yule–Simon distribution eventually became the best fit for the latest networks, i.e., for all networks sampled after a specific date $t^*$ (with the median date being December 27, 2020).

Finally, SAV links found by our approach generally agree with the links reported in other studies. In particular, the test (3) applied to the USA dataset recognized 82% of pairs listed in ref. [23] and 79% of non-trivial pairs from ref. [26] (without considering clusters of consecutive SAVs also reported in ref. [26]). It should be noted that prior studies identified much fewer linked pairs of SAVs than HELEN.

The aforementioned observations indicate that the networks inferred in this study have a sufficiently rich community structure[43] that can be analyzed and utilized to evaluate and forecast the SARS-CoV-2 evolutionary dynamics.

## Dense communities as indicators of variant emergence

We analyzed communities within coordinated substitution networks in search for evidence in support of the following hypotheses:

(H1) known VOCs/VOIs emerge as dense communities in coordinated substitution networks;

(H2) conversely, dense communities within coordinated substitution networks correspond to haplotypes with altered phenotypes;

(H3) such communities can be detected before the corresponding lineages achieve significant frequencies.

The theoretical framework for hypotheses (H1)−(H3) is established in "Model-based rationale of the proposed approach". However, the primary objective of this research is to verify these hypotheses on using empirical data. To accomplish this, we employed a dual-faceted approach. First, we performed a retrospective statistical analysis of densities of known VOCs and VOIs in coordinated substitution networks. Second, we evaluated the ability to accurately infer haplotypes with altered transmissibilities, both known and unknown, from collections of candidate dense communities. Our assessment covered several factors:

- Precision and recall of VOC/VOI detection, both on an individual country basis and jointly.
- Promptness of detection measured using so-called forecasting depth. This quantitative measure is defined as the time gap between the first variant call and the occurrence of a specific epidemiological benchmark event $b$. In this study, we used two benchmark events: (a) the variant's designation by WHO ($b$ = des) and (b) the moment its prevalence reaches 1% or, if that does not occur, when the prevalence peaks ($b$ = prev, the similar benchmark was used in ref. [3]). The value of $FD^b(h)$ can be positive or negative, thus indicating early or late prediction, respectively.
- Cumulative frequencies and prevalences of viral variant at earliest times of detection.

It is worth noting that the presence of a viral variant as a dense community does not necessarily indicate its circulation at that time. In the context of this study's model, this fact should be rather interpreted as an indication that the corresponding SAVs are linked densely enough to suggest the variant's viability. In particular, detecting the variant as a dense community in a particular country at an early time point does not necessarily mean that the variant originated there. As demonstrated below, while there are instances where this is true, more often the variants are detected earlier in countries with larger sample sizes that provide greater statistical power for inferring coordinated substitutions.

## VOCs/VOIs as communities in coordinated substitution networks

To validate hypotheses (H1) and (H3), we estimated density-based $P$ values of known VOCs and VOIs for each country and each time point using the algorithm described in "Estimation of density-based $P$ values of viral haplotypes". The algorithm produces uniform samples of connected communities of each epistatic network, and compares their densities with those of the VOCs/VOIs to calculate $P$ values. As a result, for each country and each VOC/VOI we obtained a time series of $P$ values. The series were adjusted by calculating FDR and applying the Benjamini−Hochberg procedure[44]. The resulting time series of adjusted $P$ values for the complete and truncated datasets are illustrated in Fig. 2a and Supplementary Figs. 2−31.

Our analysis of time series data showed that a significant proportion of cases exhibited variant expansion either succeeding or concurrent with a decrease in density-based $P$ values. To quantify this relationship, we employed sample cross-correlation[45] to measure the connection between $P$ values and variant prevalences throughout the growth period of the variant. We considered a range of positive and negative lags for the prevalence series in relation to $P$ value series and identified the optimal lag $l^*$ with the maximum absolute cross-correlation.

In what follows, we describe the results for the first truncated dataset; the results for the complete and second truncated datasets are in general comparable and can be found in the Supplementary Tables 1 and 2. In 74% of all test cases, we detected a non-negative optimal lag and a medium-to-strong statistically significant negative correlation between $P$ values and lagged prevalences (mean $\rho$: −0.74,

95% CI for $\rho$: [−0.97, −0.36]; mean $l^*$: 20.5, 95% CI for $l^*$: [0, 168] (in days), Supplementary Fig. 1). Focusing solely on VOCs, we observed this effect in 84% of cases (mean $\rho$: −0.72, 95% CI for $\rho$: [−0.95, −0.36]; mean $l^*$: 30.9, 95% CI for $l^*$: [0, 168] (in days)).

We defined a variant as significantly dense when its adjusted $P$ value falls below 0.05 and at least 80% of its SAVs belong to the network's giant component (the 80% threshold was selected to allow for a single AA mismatch for shortest VOCs/VOIs). For the first truncated dataset, 64% of VOCs/VOIs, analyzed separately for different countries, became significantly dense at some moment of time. This percentage increased to 93% when only considering VOCs. Moreover, the variants were identified as significantly dense at low cumulative frequencies (median value $\mu = 4 \cdot 10^{-4}$, Fig. 2d) and low prevalences ($\mu = 8 \cdot 10^{-4}$, Fig. 2e).

We assessed forecasting depths, $FD^{prev}$ and $FD^{des}$, with respect to times when the variants reached significant density. In general, VOCs/VOIs that achieved significant density tended to do so early. In particular, such variants were identified before reaching 1% prevalence in 57% of cases and before WHO designation times in 52% of cases. For early calls (i.e., given that $FD^{prev} > 0$ or $FD^{des} > 0$), the median forecasting depths were 60 and 48 days, respectively. It should be noted that forecasting depths for truncated datasets are lower, going from median $FD^{prev} = 68$ and $FD^{prev} = 66$ for the complete dataset to median $FD^{prev} = 60$ and $FD^{prev} = 35$ for the second truncated dataset (Supplementary Table 3).

In genomic surveillance, decisions are typically made based on the multitude of signals from several countries. In this context, it is important to note that all variants of concern (VOCs) and variants of interest (VOIs) have positive forecasting depths $FD^{prev}$ and $FD^{des}$ in at least one country (Fig. 2). For instance, the Omicron variant (lineage B.1.1.529.1) becomes significantly dense before its designation time and before reaching 1% prevalence in 9 countries, with forecasting depths ranging from 4 to 319 days for $FD^{des}$ and 15 to 345 days for $FD^{prev}$. The Delta variant (B.1.617.2) serves as another example of multiple early predictions, as it becomes significantly dense before its designation in 13 countries ($FD^{des} \in [15, 300]$) and before reaching 1% prevalence−in 10 countries ($FD^{prev} \in [30, 300]$).

Sample size seems to significantly impact the haplotype detection. A positive correlation exists between the number of significantly dense VOCs/VOIs and the number of sequences per country ($\rho = 0.59$, $P = 0.017$). In particular, in the United States, which has the highest number of sequences, all ten variants reached significant density.

## Inference of viral variants as dense network communities

In our analysis, we used $f$-score as a metric for comparison of inferred dense communities and known viral variants. In our context, it is defined as follows:

$$R_{t,i} = \frac{|C_t \cap V_i|}{|V_i|}, \quad P_{t,i} = \frac{|C_t \cap V_i|}{|C_t|}, \quad F_{t,i} = 2\frac{R_{t,i} \cdot P_{t,i}}{R_{t,i} + P_{t,i}}, \quad (1)$$

where $R_{t,i}$, $P_{t,i}$, and $F_{t,i}$ are the recall, precision and $f$-score for the SAVs of the variant $V_i$ found within the dense community $C_t$ at the time $t$. In what follows, we used a 80% $f$-score threshold to declare a variant detection as a dense community of a coordinated substitution network.

The most straightforward way to partially assess the validity of hypotheses (H2) and (H3) is to retrieve the densest subnetworks of coordinated substitution networks and compare them to known viral variants. This task is made easier by the fact that finding the densest subgraphs, based on our density definition, is a polynomially solvable problem (see "Inference of viral haplotypes"). The examination of the densest subnetworks indeed lends support to the hypotheses. In particular, in all three datasets every VOC emerged in at least one country as a dense community before its official designation (Fig. 3 and

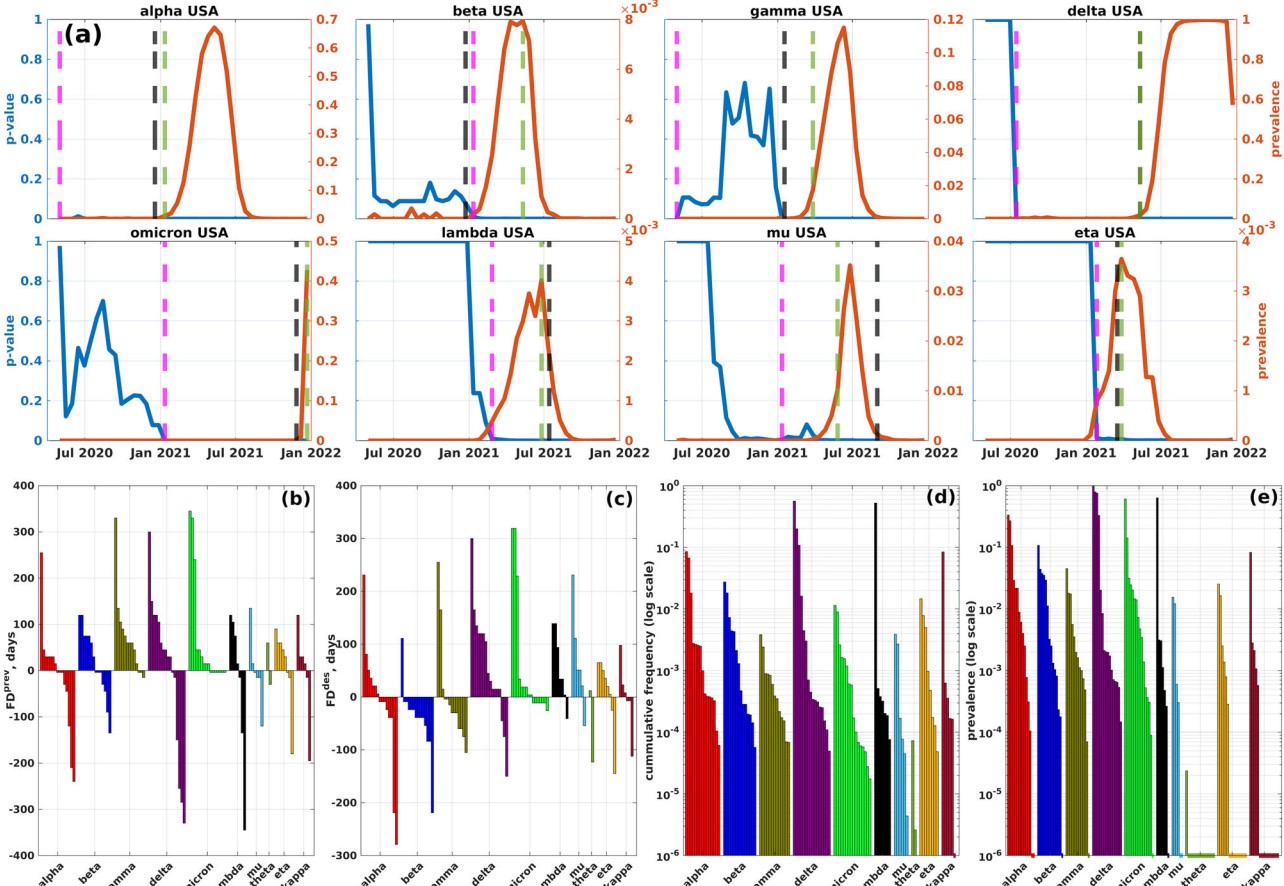

**Fig. 2 | Density-based Benjamini–Hochberg-adjusted *P* values of VOCs/VOIs (first truncated dataset). a** *P* values (blue) and prevalences (red) of 8 VOCs and VOIs in the USA coordinated substitution networks (refer to Supplementary Figs. 2–31 for all VOCs/VOIs across all countries). Black, green, and magenta lines represent the times of VOC designation, achieving 1% prevalence, and becoming significantly dense, respectively. **b**, **c** Forecasting depths (*y* axis) in relation to the 1% prevalence time and WHO designation time for each analyzed VOC/VOI across

different countries. **d**, **e** Cumulative frequencies and prevalences for VOCs/VOIs across various countries at the times when they become significantly dense (in a logarithmic scale). Dashed lines at the bottom of the plot indicate that the variants reached significant density at frequencies/prevalences of 0. For similar summaries for the complete and second truncated datasets, see Supplementary Figs. 32 and 33.

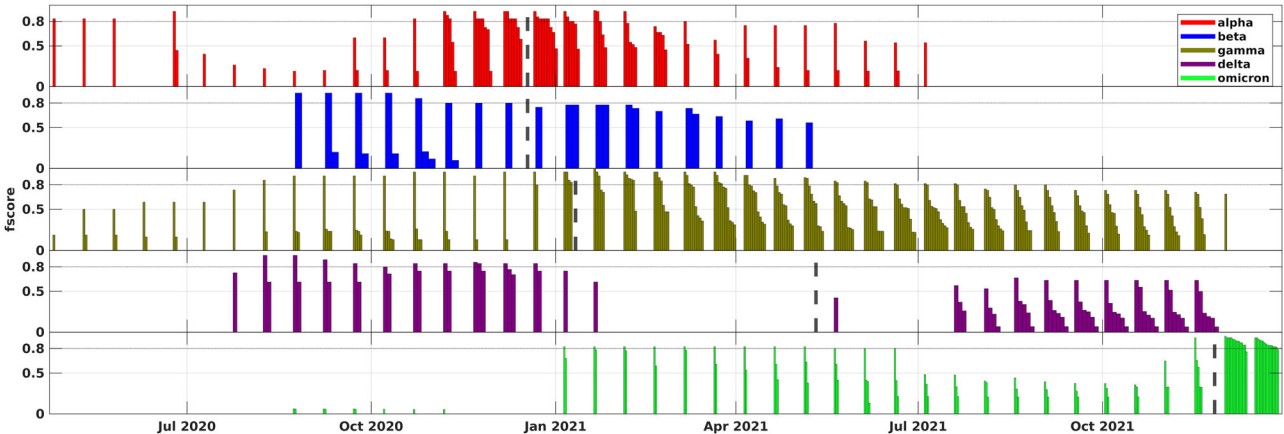

**Fig. 3 | Comparison of densest subnetworks from coordinated substitution networks (aggregated over 16 countries) with VOCs, first truncated dataset.** Similar visuals for other datasets and individual countries can be found in Supplementary Figs. 34–36. Each bar in the plot represents a specific VOC. For every

time point, the bars display the densest subgraphs of different countries that are most similar to that VOC, with the height of the bars indicating the corresponding *f*-scores. Dashed lines highlight the moments when the WHO designated the VOCs.

Supplementary Figs. 37 and 38). All VOCs were also detected before reaching the 1% prevalence, except for the Beta variant, that was detected as soon as it reached the 1% mark. The detailed results of the densest subnetwork analysis are reported in the Supplement.

However, a more advanced algorithmic approach is essential for a comprehensive early detection framework, as well as for stronger hypotheses confirmation. Indeed, generally, only a single densest subnetwork can be constructed per time point, even though multiple haplotypes with altered phenotypes might coexist at each specific moment. As a result, for example, no VOI was detected as a densest subnetwork. In addition, we observed that, as coordinated substitution networks become denser over time, the densest subnetworks expand and may ultimately encompass several haplotypes, leading to decreased variant inference accuracy.

To overcome these problems, we developed HELEN—a more complex algorithm for inferring viral haplotypes as dense network communities ("Inference of viral haplotypes"). Briefly, HELEN generates a pool of distinct dense subnetworks of varied sizes, partitions them into clusters, and assembles a haplotype from each cluster using graph-theoretical techniques. For every assembled haplotype, the algorithm also returns its *support* defined as the percentage of candidate subnetworks corresponding to that haplotype.

Sensitivity outcomes for the proposed algorithm are illustrated in Fig. 4, Supplementary Figs. 44 and 45 and Table 1. The numbers below are summary statistics ranges for three analyzed datasets.

The recall of known VOCs/VOIs can be assessed in two ways:

- When countries are assessed individually, the summary statistics can be reported as an aggregated recall $R = \frac{1}{nm}\sum_{i=1}^{n}\sum_{j=1}^{m}\chi_{i,j}$. Here, $n$ is the number of countries, $m$ is the number of viral variants, and $\chi_{ij}$ is a binary indicator, set to 1 if variant $j$ is identified in country $i$. Under this approach, HELEN exhibits an aggregated recall rate between 50 and 53% for three datasets. This is reasonable, especially given the varying prevalence of VOIs across countries. Moreover, focusing solely on VOCs, the aggregated recall increases to 90–93%. These numbers represent a 2- to 2.5-fold improvement over the densest subgraph-based method.

- From a genomic surveillance standpoint, it is also meaningful to assess recall based on the combined signal from all countries. Under this approach, HELEN detected 8–9 out of 10 examined variants in at least one country, failing to detect the Theta variant in all datasets and the Mu variant in the complete dataset. It is worth mentioning that for the latter case, communities with as high as 0.75 identity were detected several times, narrowly missing our pre-defined threshold. All VOCs were found in 12–16 of the 16 analyzed countries, whereas detected VOIs ranged from being present in 1–6 countries.

A significant proportion of these detections occurred early. Specifically, 44–47% of the earliest VOCs/VOIs detections happened before they reached a 1% prevalence and 40–45% were first detected before their WHO designation. Upon first detection, the median variant frequency lay between $3.37 \cdot 10^{-4}$ and $3.99 \cdot 10^{-4}$, while the median variant prevalence ranged from $1.21 \cdot 10^{-3}$ and $1.61 \cdot 10^{-3}$. Again, these values signify three- to fourfold improvement over the densest subgraph-based method with respect to the frequency, and 11–15-fold improvement with respect to the prevalence.

In terms of forecasting depths, 7–9 of the 10 variants exhibited non-negative values of $FD^{prev}$, and 8–9 out of 10 had non-negative $FD^{des}$ values in at least one country. However, the forecasting depths vary among the three datasets. The median depth $FD^{prev}$ is 60 days for the complete dataset, which is higher than the 45 days for the truncated datasets. Likewise, $FD^{des}$ is 67 days for the complete dataset, decreasing to 56 for the first truncated dataset and 36 days for the second. Such variation is expected given the definitions of the datasets.

Regarding specific variants, all VOCs were detected early in all datasets. The maximum forecasting depths differ noticeably among datasets, but they are generally reasonably high. For the complete dataset, the maximum $FD^{prev}$ values span from 120 days (for Beta) to 360 days (for Delta). In contrast, for the second truncated dataset, these values range from 30 days (Omicron) to 285 days (Delta). Similarly, the maximum $FD^{des}$ values range from 111 days (Beta) to 360 days (Delta) in the complete dataset, and from 4 days (Omicron) to 285 days (Delta) in the second truncated dataset (see Fig. 4 and Supplementary Figs. 44 and 45).

The VOIs, while generally showing more decent forecasting depths, had early identifications for Lambda, Mu, Eta, and Kappa variants. These were detected between 5 and 124 days before their WHO designation and 0–75 days before they reached a 1% prevalence, as seen in Fig. 4 and Supplementary Figs. 44 and 45. Notably, some forecasting depths actually increased for the truncated datasets. Taking the Eta variant as an example, the maximum $FD^{prev}$ rose from 30 days (in the complete dataset) to 60 days (in the second truncated dataset). Similarly, the maximum $FD^{des}$ shifted from 5 days to 35 days. This phenomenon, together with the detection of the Mu variants only in truncated datasets, can be associated with the opposite effect observed for the VOCs: without the dense communities related to VOCs at certain times, the algorithm could detect VOI-associated communities earlier.

Similar to the case with significantly dense subgraphs, sample sizes and geographic diversity influence variant detection. A medium-to-strong positive correlation was observed between the number of sequences per country and the number of variants with positive forecasting depths (Table 1). Some of the earliest forecasts, although not all, were made in the countries of origin for specific variants: notably, Beta, Gamma, and Lambda variants were detected in South Africa, Brazil, and Peru (Supplementary Figs. 41–43). On the other hand, failure to detect Theta variant can be attributed to the fact that 80% of theta cases were observed in the Philippines, a country not included in our analysis due to the smaller sample size. Haplotype size does not significantly affect the accuracy of detection, as the correlation between VOC/VOI numbers of SAVs and average $f$-values at detection was not statistically significant ($\rho = 0.17$, $P = 0.65$).

To assess the precision of HELEN, it is important to consider that the true positive network communities identified by the algorithm might not only correspond to known VOCs/VOIs but also to variants exhibiting increased transmissibility that failed to become VOC/VOI due to factors such as genetic drift or containment through public health measures before achieving a high global prevalence. Consequently, we classify a haplotype $\nu$ identified by HELEN at a specific time as spreading, if $\nu$ is a known VOC/VOI or if the prevalence of variants highly similar to $\nu$ has increased or will increase by a factor of 10 in the past or future. Note that a similar fold-based criterion was employed to define spreading mutations in ref. 3. A variant $\nu'$ is considered highly similar to $\nu$ if it contains at least 80% of $\nu$'s SAVs; this definition encompasses variants genetically close to $\nu$ and their descendants.

We measure precision using the matching similarity metric, denoted as $A_{I \to S}$. This metric evaluates the agreement between inferred haplotypes ($I$) and spreading haplotypes ($S$) by taking into account haplotype support as a proxy for haplotype call confidence and measuring the extent to which inferred haplotypes, weighted by their support ($\sigma_i: i \in I$), are matched by their nearest spreading haplotypes. Formally, the matching similarity is the average $f$-score for inferred haplotypes in relation to their closest spreading haplotypes:

$$A_{I \to S} = \sum_{i \in I} \sigma_i \max_{s \in S} f_{i,s} \qquad (2)$$

A similar measure, in the reverse form of a matching error, was used, e.g., in ref. 36.

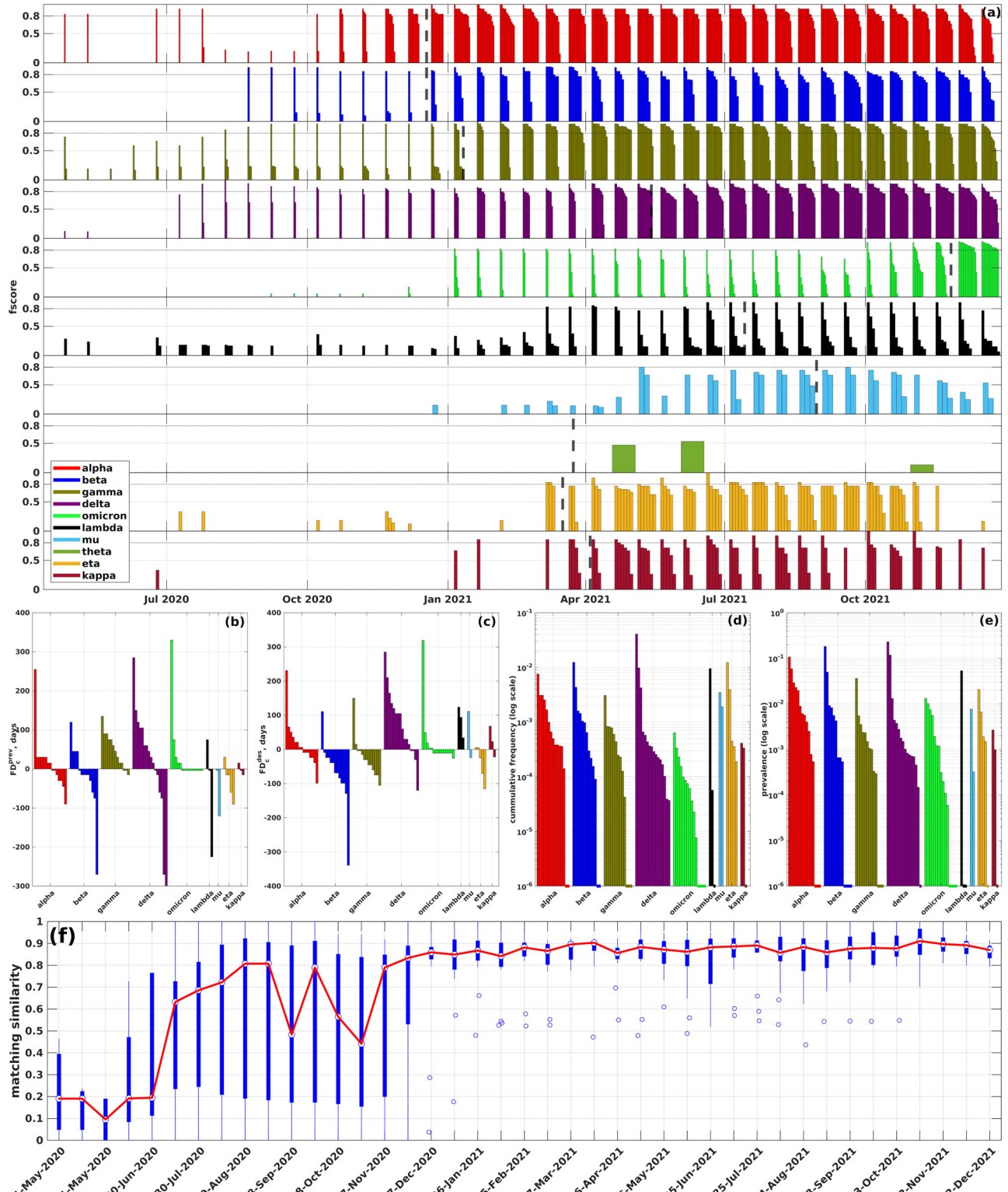

**Fig. 4 | Analysis of inferred haplotypes. a** Summary of comparison between VOCs/VOIs and inferred haplotypes (first truncated dataset, the results for other datasets are depicted on Supplementary Figs. 44 and 45). Each bar plot depicts the comparison results for a particular VOC/VOI; at each time point, bars correspond to inferred haplotypes from different countries closest to that VOC, and the bar heights are equal to the respective *f*-scores. Colored dashed lines mark times when the VOCs were designated by WHO. **b**, **c** Forecasting depths (*y* axis) with respect to the 1% prevalence time and WHO designation time for each analyzed VOCs/VOIs over different countries. **d**, **e** Cumulative frequencies and prevalences of VOCs/VOIs over different countries at first variant call times (in logarithmic scale). Dashed lines at the bottom of the plot signify that the corresponding variants were detected at cumulative frequencies or prevalences 0. **f** Precision of haplotype inference. Blue box plot depicts summary statistics of matching similarity of *n* = 16 countries over *T* = 21 time points. The bottom and top of each box are the 25th and 75th percentiles, whiskers represent minimum and maximum values, white dot is a median. Red plot depicts the dynamics of median matching similarity over time.

**Table 1 | Summary statistics for VOC/VOI recall by HELEN**

| | Complete | 1st truncated | 2nd truncated |
|---|---|---|---|
| VOCs/VOIs identified in at least one country | 8/10 | 9/10 | 9/10 |
| Number of countries where VOCs (VOIs) were detected | 13–15 (1–4) | 13–16 (2–5) | 13–16 (2–6) |
| Aggregated recall for VOCs/VOIs (VOCs only). | 50% (90%) | 53% (90%) | 55% (93%) |
| Percentage of earliest VOCs/VOIs detections with $FD^{prev} \geq 0$ | 45% | 47% | 44% |
| Percentage of earliest VOCs/VOIs detections with $FD^{des} \geq 0$ | 45% | 42% | 40% |
| Median cumulative frequency at first detection | $3.99 \cdot 10^{-4}$ | $3.37 \cdot 10^{-4}$ | $3.77 \cdot 10^{-4}$ |
| Median prevalence at first detection | $1.48 \cdot 10^{-3}$ | $1.21 \cdot 10^{-3}$ | $1.61 \cdot 10^{-3}$ |
| VOCs/VOIs (VOCs) with $FD^{prev} \geq 0$ in at least one country | 7/10 (5/5) | 9/10 (5/5) | 9/10 (5/5) |
| VOCs/VOIs (VOCs) with $FD^{des} \geq 0$ in at least one country | 8/10 (5/5) | 9/10 (5/5) | 9/10 (5/5) |
| Median $FD^{prev}$ for early calls, days | 60 | 45 | 45 |
| Median $FD^{des}$ for early calls, days | 67 | 56 | 36 |
| Linear correlation and *P* value (two-sided Student's *t* test) for the number of sequences per country and the number of variants with $FD^{des} > 0$ | 0.33 (0.21) | 0.54 (0.031) | 0.38 (0.15) |
| Linear correlation and *P* value (two-sided Student's *t* test) for the number of sequences per country and the number of variants with $FD^{prev} > 0$ | 0.72 (0.0016) | 0.70 (0.0025) | 0.73 (0.0013) |

The summary statistics for matching similarity at each time point across different countries is summarized by Fig. 4 and Supplementary Figs. 44 and 45. The general trend is the precision growth over time during the first year of the pandemic followed by the relatively steady state during the second year. For example, for the first truncated dataset (Fig. 4f) HELEN initially achieved a median matching similarity above 80% in August, 2020, and stayed above 85% from December 2020. Initially, there was a considerable variation in matching accuracy among countries, but it noticeably declined by early 2021. These observations can be associated with the density dynamics of coordinated substitution networks in different countries, whereas the precision increases as more epistatically linked SAVs are identified.

Finally, we compared the accuracy results of HELEN with those based on findings of ref. 26, that similarly identified clusters of concordantly evolving spike protein sites in coordinated substitution networks using an alternative approach. The comparison focused on data aggregated up to September 7, 2021, to match the dataset used in ref. 26. As above, to measure recall, we estimated VOC/VOI *f*-scores in relation to the closest inferred clusters, while for precision we, conversely, calculated *f*-scores of inferred clusters in relation to the nearest VOCs/VOIs. As ref. 26 does not report a confidence measure to calculate the summary statistics akin to matching similarity, we focused on HELEN clusters with over 1% support. The comparison demonstrates that HELEN achieves higher recall and precision (Supplementary Fig. 47).

### Running time and scalability
The computational methods employed in this study are reasonably efficient and scale to millions of sequences. For instance, for the US dataset analyzed at the time point $t = 37$, which consists of ~1.66 · 10^6 sequences, constructing the coordinated substitution network took ~1 h, estimating the *P* values of 10 VOCs/VOIs took ~1.8 h, and inferring viral haplotypes took ~38.6 h. HELEN is developed using Matlab R2023a and Gurobi 10.0.3 and all the computations were carried out on a workstation equipped with a 3 GHz Intel Xeon E5 CPU and 64GB of RAM.

### Discussion
This study explores the hypothesis that viral variants with higher transmissibility can be associated with dense communities in coordinated substitution networks. Specifically, we investigated this idea in the context of SARS-CoV-2 spike protein genomic variants and found strong support for it. Our results indicate that network density can serve as a dependable indicator for the timely detection or prediction of emerging SARS-CoV-2 variants. As a result, we proposed an accurate, interpretable, and scalable method that can anticipate emerging SARS-CoV-2 haplotypes several months in advance, leading to early detection and improved forecasting.

These results were obtained using a synthetic approach that combines methods from statistics, combinatorial optimization, and population genetics. Firstly, we employed a sensitive statistical test that relies on a quasispecies population genetics model to identify linked pairs of SAVs that are jointly observed more often than expected if the corresponding 2-haplotype is inviable. This method allowed us to construct coordinated substitution networks with rich community structures, providing a foundation for meaningful network-based inference. Secondly, we validated our hypothesis by estimating network density-based *P* values of SARS-CoV-2 haplotypes. This allowed us to identify haplotypes with low *P* values as potential variants of concern and demonstrate that known VOCs achieve low *P* values significantly earlier than they reach frequencies high enough to be detected using conventional methods. Lastly, we utilized these findings to design an algorithm for the early detection of viral variants that identifies dense communities of SAV alleles and combines them into haplotypes. We demonstrate the efficacy of this algorithm by retrospectively identifying known VOCs and VOIs with high accuracy up to several months before they reached high prevalence and were designated by the WHO.

Compared to traditional phylogenetic lineage tracing, the proposed methodology offers several advantages. In particular, it can detect viral variants as dense communities at very low frequencies or even when actual variant sequences are not sampled—the latter is possible when there are sufficiently many well-covered variant's SAV pairs. This feature is naturally inherited from our prior methods[36,39] for reconstructing intra-host viral populations from noisy NGS data, which have demonstrated the ability to accurately detect viral haplotypes with frequencies as low as the level of sequencing noise. In addition, the computational complexity of most intensive steps of network-based methods is a function of the genome length rather than the sequence number. For SARS-CoV-2 data, the number of available sequences in GISAID is up to 4 orders of magnitude larger than the number of amino acid positions in the SARS-CoV-2 s-gene (~1.5 · 10^7 sequences versus 1.27 · 10^3 amino acid positions). This feature makes the proposed algorithms considerably more scalable than phylogenetic methods.

It is important to note that there are limitations to this study, as the comprehensive forecasting of viral evolution is inherently an intractable problem. While the proposed methods have shown

promising early detection results, caution should be exercised when interpreting them. First of all, our findings by no means suggest that viral evolution is a deterministic process that can be predicted using mechanistic models. Instead, they demonstrate how to identify potential evolutionary trajectories among exponentially many possibilities. These trajectories can guide further investigation and prioritization of functional screening. Nonetheless, the number of these trajectories could be substantial. For instance, in the idealized model presented in "Model-based rationale of the proposed approach", the number of predictable high-fitness variants corresponds to the number of maximal cliques within an epistatic network. Although this is typically much smaller than the overall number of potential genotypes, in the worst case it may still be exponential[46].

Moreover, the GISAID data used in our study encompasses sequences obtained under different conditions by a variety of laboratories worldwide. Consequently, despite GISAID efforts to maintain consistent quality control, there may still be variations in the reliability of the sequences and their associated metadata. Specifically, there are concerns that the complete dataset might be less trustworthy than its truncated counterparts, potentially containing mislabelled or contaminated data. Nonetheless, we opted to analyze this dataset to ensure thoroughness and to highlight the sensitivity of the proposed methodology, irregardless of data specifics. It is imperative, however, to approach the forecasting depths of this dataset with a degree of skepticism. Primarily, our results serve as a testament to the methodology's capacity to detect rare genotypes with altered phenotypes in a genomic sample, irrespective of the provenance of these genotypes. Exploration of the origins of SARS-CoV-2 VOCs and VOIs is beyond the scope of this study.

Next, the links between SAVs identified by HELEN represent putative or potential positive epistatic interactions[23], and their primary purpose is to serve as features for our prediction model. These links should be viewed as a statistical ensemble rather than individually, with our findings suggesting that haplotypes with altered phenotypes exhibit a significantly higher number of potential epistatic pairs compared to background haplotypes. Consequently, research focused on examining the biological mechanisms of specific SARS-CoV-2 epistatic interactions should incorporate more comprehensive structural data.

The utilized coordinated substitution/epistasis model is another limitation of this study as it only considers the interactions between SAV pairs, thus reflecting pairwise or second-order epistasis. Although combinations of mutations can have more complex fitness effects involving higher orders of epistasis[47], this model is justifiable for several computational reasons. First, it is the minimal model that enables the detection of multiple overlapping haplotypes, which is an improvement over the mutation independence assumption used in other studies[3] that, in general, only allow ranking and prioritization of mutations. Secondly, $k$-haplotypes with $k \geq 3$ may not have sufficiently high frequencies to be detected, thereby affecting the method's predictive power. In contrast, pairs are always covered by more sequences and can be detected earlier. Lastly, accounting for higher-order combinations of mutations can increase the computational complexity of the problem while the second-order model remains computationally tractable.

Finally, our method is based solely on genomic data, and its effectiveness could be enhanced by incorporating epidemiological and structural biology data and models. In addition, our results highlight the significance of robust and diverse sampling practices, as early detections were predominantly made in countries with larger sample sizes, and some variants were only detected early in their countries of origin.

We believe that the methodology proposed in this study is not limited to SARS-CoV-2 and can be extended to other pathogens. The high sensitivity of HELEN positions it as an effective tool for forecasting emerging and detecting circulating strains of pandemic viruses, including HIV, Hepatitis C, and Influenza. This capability is particularly valuable in the context of seasonal vaccine development, where accurate and timely forecasts can play a crucial role in the selection of strains for vaccine formulation.

## Methods
The major goal of this study is to develop and validate a methodology that, given viral sequences sampled at several time points, infers potentially emerging viral haplotypes by analyzing dense communities of coordinated substitution networks. To achieve it, this section is organized as follows. First, we provide a theoretical justification of the proposed approach by considering an idealized model of an evolving population with given fitness landscape and epistatic network ("Model-based rationale of the proposed approach"). As epistatic networks are not directly observable, "Inference of coordinated substitution networks" outlines the methodology to infer them from sequencing data. "Estimation of density-based $P$ values of viral haplotypes" describes our approach to validate the statistical significance of associations between known VOCs/VOIs and dense communities in inferred epistatic networks. Finally, "Inference of viral haplotypes" presents the algorithmic framework to de novo infer emerging viral variants.

### Model-based rationale of the proposed approach
The major idea of this study is to predict emerging viral variants as dense communities in epistatic networks. This idea can be partially substantiated by the following simple combinatorial population genetics model assuming that the basic mutational mechanism consists of random point mutations.

Consider a population of haploid genotypes $\mathcal{P} = \{g_1, \ldots, g_n\}$ with a fixed length $L$ and two potential allelic states 0 and 1 at each locus, where 0 stands for the reference allele and 1 stands for an alternative allele. Each genotype is thus represented as a binary sequence, and all possible $2^L$ genotypes form a sequence space represented as $L$-dimensional hypercube $\mathcal{H}$[48], i.e., a graph whose vertices are $0-1$ sequences of length $L$, and two vertices are adjacent whenever they differ in a single coordinate.

Each genotype $g_i$ is assigned the fitness $f_i$ – a real number that serves as a quantitative measure of its reproductive capacity[49,50]. The function mapping genotypes into the set of their fitness values is referred to as a fitness landscape[49].

In our model, the genotypes are subject to negative epistasis[51,52]. Following refs. [50,53], it is defined as the statistical effect where the combined effect of mutations at two specific loci leads to a lower fitness than if these mutations occurred independently, i.e., $f_{11} < f_{10} + f_{01} - f_{00}$, where $f_{ij}$, $i, j = 0, 1$ are expected fitnesses of genotypes with allelic states $(i, j)$ at the loci. Similarly, positive epistasis occurs when the combined effect of multiple mutations results in a higher than expected fitness, i.e., $f_{11} > f_{10} + f_{01} - f_{00}$[50,53]. In our case, negative epistasis is assumed to render the corresponding genomes non-viable or evolutionary non-competitive ($f_{11} \leq 0$). Epistatic interactions can be represented by the coordinated substitution network $\mathcal{G}$ (Fig. 5a), where vertices correspond to loci, and two vertices are adjacent when a 2-haplotype $(1, 1)$ at the respective loci is viable (i.e., the loci are not under negative epistasis).

Epistasis has been proposed to constrain the selective accessibility of genomic variants and restrict potential evolutionary trajectories of a population[51,54]. This effect can be described in graph-theoretical terms as follows. Viable genotypes (i.e., genotypes with positive fitness values) constitute a subgraph of the hypercube $\mathcal{H}$, referred to as the viable space. In the model under consideration, a genotype is deemed viable if all its alternative alleles are pairwise adjacent in the network $\mathcal{G}$ i.e., create a clique within $\mathcal{G}$ (Fig. 5b). As a result, each maximal by inclusion clique $C$ of $\mathcal{G}$ (i.e., a clique that is not contained in another clique) generates a complete sub-hypercube

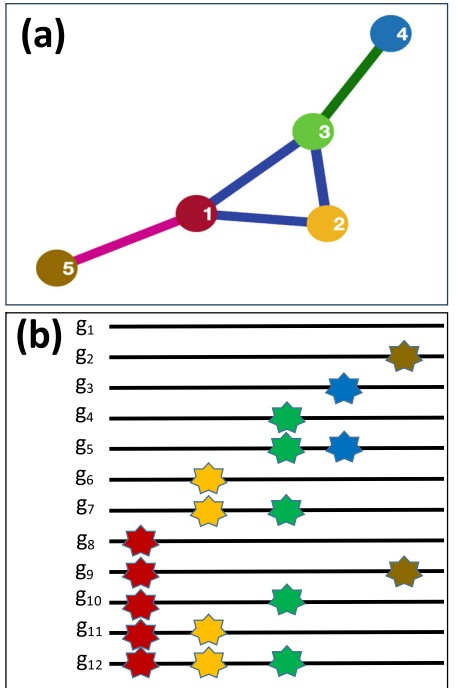

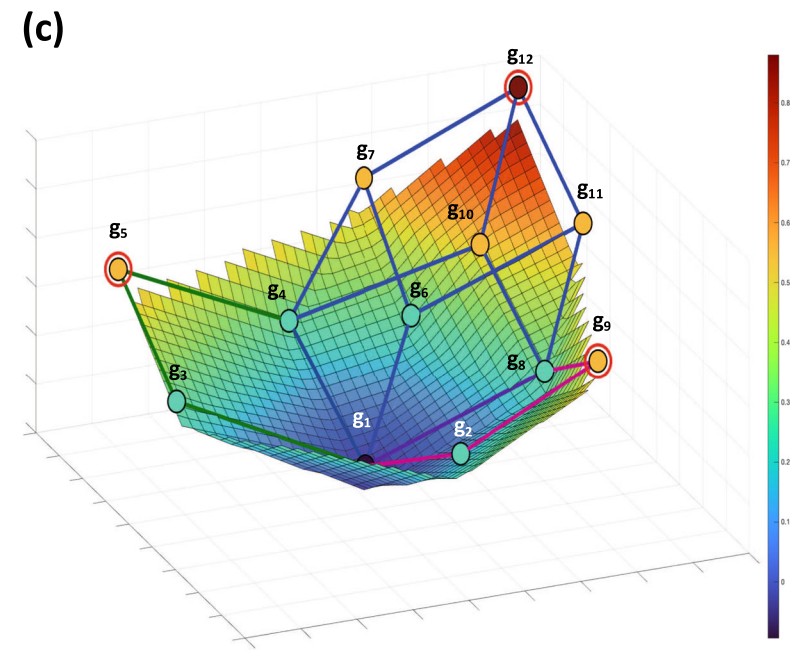

**Fig. 5 | The model of an epistatically-constrained sequence space and fitness landscape. a** The epistatic network $\mathcal{G}$. Edges of inclusion-maximal cliques are displayed in blue, green and purple. **b** Genotypes that are viable under the constraints imposed by the epistatic networks. Stars represent 1-alleles, colors denote loci. **c** The viable space is depicted alongside the corresponding fitness landscape. For better visualization, as is customary in the literature[70], the fitness landscape is depicted as a continuous surface. Surface and vertex colors represent fitness values on a scale from blue (low fitness) to red (high fitness). Sub-hypercubes corresponding to three maximal cliques of the epistatic network $\mathcal{G}$ are highlighted in blue, green, and purple, respectively, with edges belonging to two sub-hypercubes colored in intermediate shades. The circled vertices represent local maximums within each sub-hypercube. For example, all minor alleles of the genotypes $g_4, g_6, g_7, g_8, g_{10}, g_{11}$, and $g_{12}$ are situated at loci 1, 2, or 3. These loci form a clique of the epistatic network, while these genotypes, together with the wild-type genotype $g_0$, form a 3-dimensional sub-hypercube of the sequence space (highlighted in black in (**c**)). The genotype $g_{12}$ has the maximum fitness within this sub-hypercube.

$\mathcal{H}(C)$ in the viable space; this sub-hypercube is a projection of genotype vectors onto the subspace formed by loci from $C$. Thus, decomposing the epistatic network into maximal cliques yields a partition of the viable space into sub-hypercubes. This partition defines a set of restricted evolutionary trajectories that the population could potentially explore.

More specifically, within each sub-hypercube $\mathcal{H}(C)$, only additive and positive epistatic fitness effects can be present. Therefore, evolutionary trajectories within $\mathcal{H}(C)$ will eventually accumulate all mutations in $C$ and converge to the genotype $g_C$ with the maximum fitness within $\mathcal{H}(C)$ that contains all alternative alleles of the clique $C$ (Fig. 5c). Overall, the proposed model indicates that any evolutionary trajectory within the entire viable space will ultimately converge to a genotype determined by one of the maximal cliques in the epistatic network.

In practical settings, epistatic networks are not directly observable. Therefore, in accordance with[23,26,34], we approximate them using coordinated substitution networks, which are statistically inferred from genomic data. Since the inferred networks may not encompass all true links, we consider dense subgraphs rather than cliques.

### Inference of coordinated substitution networks

Consider a population consisting of $N$ haploid genotypes of length $L$ whose observed abundances change over time points $t = 1, ..., T$. For a pair of distinct loci $u, v \in \{1, ..., L\}$ we consider 4 possible 2-haplotypes $(i, j) \in \{(0, 0), (0, 1), (1, 0), (1, 1)\}$, where 0 and 1 are reference and alternative alleles in $u$ and $v$ respectively. Let also $O_{ij}^t$ be an observed count of 2-haplotypes $(i, j)$, $i, j = 0, 1$, at a time point $t \in \{1, ..., T\}$.

We define a coordinated substitution network at the time point $t$ as a graph $\mathcal{G}_t$ with nodes representing SAVs, and two nodes being adjacent whenever the corresponding non-reference alleles are simultaneously observed more frequently than expected by chance. Formally, SAVs at positions $u$ and $v$ are adjacent in $\mathcal{G}_t$ (or linked), when the following inequality holds:

$$1 - \sum_{i=0}^{O_{11}^t - 1} \binom{N}{i} \left( \frac{O_{10}^t \cdot O_{01}^t}{O_{00}^t \cdot N} \right)^i \left( 1 - \frac{O_{10}^t \cdot O_{01}^t}{O_{00}^t \cdot N} \right)^{N-i} \leq \frac{\rho}{\binom{L}{2}}, \quad (3)$$

where $\rho$ is a pre-defined $P$ value (in this study, we used $\rho = 0.05$).

In the remaining part of this subsection, we provide a justification for the formula (3). We suppose that viral evolution is driven by mutation and selection, where (a) each 2-haplotype $(i, j)$ has fitness $f_{ij}$; (b) each transition (mutation) from the allele $k$ to the allele $l$ at the position $u$ (resp. $v$) happens with probability $q_{kl}^u$ (resp. $q_{kl}^v$). Thus, expected 2-haplotype counts $E_{ij}^t$ can be described by the quasispecies model[55] (or mutation-selection balance model in the classical population genetics terms[56]) in the following form:

$$E_{ij}^t = \sum_{k, l \in \{0, 1\}} f_{kl} q_{kl}^u q_{lj}^v E_{kl}^{t-1} \quad (4)$$

Mutation probabilities per genomic position per year of most viruses have orders of magnitude between $10^{-3}$ and $10^{-5}$[57]. Thus, we can assume that for time intervals considered in this study, the non-negative probability of allelic change is smaller than the probability of no-change, i.e.,

$$0 < q_{ij}^u < q_{ii}^u, \ 0 < q_{ij}^v < q_{ii}^v, \ i, j \in \{0, 1\} \ i \neq j \quad (5)$$

We can use the model (4) to decide whether the 2-haplotype $(1,1)$ is viable or its observed appearances can be plausibly explained by random mutations. The corresponding test is based on the following fact:

**Theorem 1** Suppose that the 2-haplotype $(1,1)$ is not viable, i.e., $f_{11} = 0$. Then

$$E_{11}^t \leq \frac{E_{01}^t \cdot E_{10}^t}{E_{00}^t} \qquad (6)$$

***Proof*** The proof follows the same lines as the proof in ref. 39. Given that $f_{11} = 0$, we have

$$
\begin{aligned}
E_{00}^t \cdot E_{11}^t &= \left( \sum_{k,l=0,1} f_{kl} q_{k0}^u q_{l0}^v E_{kl}^{t-1} \right) \left( \sum_{k,l=0,1} f_{kl} q_{k1}^u q_{l1}^v E_{kl}^{t-1} \right) \\
&= q_{00}^u q_{00}^v q_{01}^u q_{01}^v (f_{00} E_{00}^{t-1})^2 + q_{10}^u q_{00}^v q_{11}^u q_{01}^v (f_{10} E_{10}^{t-1})^2 \\
&\quad + q_{00}^u q_{10}^v q_{01}^u q_{11}^v (f_{01} E_{01}^{t-1})^2 + \\
&\quad + (q_{00}^u q_{00}^v q_{11}^u q_{01}^v + q_{00}^u q_{10}^v q_{01}^u q_{01}^v) f_{00} f_{01} E_{00}^{t-1} E_{01}^{t-1} + \\
&\quad + (q_{00}^u q_{00}^v q_{11}^u q_{01}^v + q_{10}^u q_{00}^v q_{01}^u q_{01}^v) f_{00} f_{10} E_{00}^{t-1} E_{10}^{t-1} + \\
&\quad + (q_{00}^u q_{10}^v q_{11}^u q_{01}^v + q_{10}^u q_{00}^v q_{01}^u q_{11}^v) f_{01} f_{10} E_{01}^{t-1} E_{10}^{t-1}
\end{aligned}
\qquad (7)
$$

and

$$
\begin{aligned}
E_{01}^t \cdot E_{10}^t &= \left( \sum_{k,l=0,1} f_{kl} q_{k0}^u q_{l1}^v E_{kl}^{t-1} \right) \left( \sum_{k,l=0,1} f_{kl} q_{k1}^u q_{l0}^v E_{kl}^{t-1} \right) \\
&= q_{00}^u q_{01}^v q_{01}^u q_{00}^v (f_{00} E_{00}^{t-1})^2 + q_{10}^u q_{01}^v q_{11}^u q_{00}^v (f_{10} E_{10}^{t-1})^2 \\
&\quad + q_{00}^u q_{11}^v q_{01}^u q_{10}^v (f_{01} E_{01}^{t-1})^2 + \\
&\quad + (q_{00}^u q_{01}^v q_{01}^u q_{10}^v + q_{00}^u q_{11}^v q_{01}^u q_{00}^v) f_{00} f_{01} E_{00}^{t-1} E_{01}^{t-1} + \\
&\quad + (q_{00}^u q_{01}^v q_{11}^u q_{00}^v + q_{10}^u q_{01}^v q_{01}^u q_{00}^v) f_{00} f_{10} E_{00}^{t-1} E_{10}^{t-1} + \\
&\quad + (q_{00}^u q_{11}^v q_{11}^u q_{00}^v + q_{10}^u q_{01}^v q_{01}^u q_{10}^v) f_{01} f_{10} E_{01}^{t-1} E_{10}^{t-1}
\end{aligned}
\qquad (8)
$$

It is easy to see that the terms in Eqs. (7) and (8) except for the last ones are equal. Thus we have

$$
\begin{aligned}
E_{01}^t \cdot E_{10}^t - E_{00}^t \cdot E_{11}^t &= \\
&= (q_{00}^u q_{11}^v q_{11}^u q_{00}^v + q_{10}^u q_{01}^v q_{01}^u q_{10}^v - q_{00}^u q_{10}^v q_{11}^u q_{01}^v \\
&\quad - q_{10}^u q_{00}^v q_{01}^u q_{11}^v) f_{01} f_{10} E_{01}^{t-1} E_{10}^{t-1} = \\
&= \left( 1 - \frac{q_{01}^u q_{10}^u}{q_{00}^u q_{11}^u} \right) \left( 1 - \frac{q_{01}^v q_{10}^v}{q_{00}^v q_{11}^v} \right) q_{00}^u q_{11}^v q_{11}^u q_{00}^v f_{01} f_{10} E_{01}^{t-1} E_{10}^{t-1} \geq 0,
\end{aligned}
\qquad (9)
$$

where the last inequality follows from (5). Thus, the inequality (6) holds.

Using Theorem 1, we can evaluate the likelihood of the event that a large number of genomes contain 2-haplotype $(1,1)$ given that this 2-haplotype is not viable. Considering the density of sampling and the number of SARS-CoV-2 genomes analyzed in this study, we assume that observed and expected numbers of 2-haplotypes are close to each other. Let $q$ is the probability of observing a genome containing 2-haplotypes $(1,1)$ among $N$ genomes given that $f_{11} = 0$. Following ref. 36, we model the count of such genomes, $X$, with a binomial distribution $B(N,q)$. The probability that $X \geq O_{11}^t$ is:

$$p(X \geq O_{11}^t | f_{11} = 0) = 1 - F_X(O_{11}^t - 1 | N, q) = 1 - \sum_{i=0}^{O_{11}^t - 1} \binom{N}{i} q^i (1-q)^{N-i}, \qquad (10)$$

where $F_X$ is the binomial cumulative distribution function. Theorem 1 implies an upper bound for $q$: $q \leq p = \frac{O_{10}^t \cdot O_{01}^t}{O_{00}^t \cdot N}$. Therefore

$$p(X \geq O_{11}^t | f_{11} = 0) = 1 - F_X(O_{11}^t - 1 | N, q) \leq 1 - F_X(O_{11}^t - 1 | N, p). \qquad (11)$$

We consider SAVs at positions $u$ and $v$ linked when the probability $p(X \geq O_{11}^t | f_{11} = 0)$ is sufficiently low, which is guaranteed when its upper bound in Eq. (11) is sufficiently low, i.e.,

$$p(X \geq O_{11}^t | f_{11} = 0) \leq 1 - F_X(O_{11}^t - 1 | N, p) \leq \frac{\rho}{\binom{L}{2}}, \qquad (12)$$

where $\rho$ is a chosen significance level, and the denominator $\binom{L}{2}$ is a Bonferroni correction. The latter is used to account for multiple comparisons between $\binom{L}{2}$ pairs of SAVs being tested for linkage. This leads to the formula (10).

### Estimation of density-based *P* values of viral haplotypes

We hypothesize that viral haplotypes corresponding to potential VOCs and VOIs form dense subgraphs of coordinated substitution networks. Below we describe the method used to statistically verify this hypothesis.

In what follows, we will use the standard graph-theoretical notation: $V(\mathcal{G})$ and $E(\mathcal{G})$ are the sets of vertices and edges of the graph $\mathcal{G}$, respectively; $N_\mathcal{G}(v)$ is the set of neighbors of a vertex $v$ in $\mathcal{G}$; the subgraph of $G$ induced by a subset $S$ is denoted by $\mathcal{G}[S]$.

We use the statistical test (12) to construct coordinated substitution networks $\mathcal{G}_t$ for different time points $t$ using SARS-CoV-2 sequences sampled before or at the time $t$. These networks have the same set of vertices but different sets of edges. A viral haplotype thus can be associated with a subset of vertices $H \subseteq V(\mathcal{G}_t)$ of a network $\mathcal{G}_t$. The density of a haplotype $H$ is thus defined as the density of the subgraph of $\mathcal{G}_t$ induced by $H$, i.e.,

$$d_{\mathcal{G}_t}(H) = \frac{|E(\mathcal{G}_t[H])|}{|H|} \qquad (13)$$

We estimate the statistical significance of our hypothesis by producing density-based *P* values of known VOC and VOI haplotypes $H$. Given the subgraph sample $\mathcal{S}^* = \{S_1, \ldots, S_{|\mathcal{S}^*|}\}$, *P* value of a haplotype $H$ in the network $\mathcal{G}_t$ is defined as

$$p_{\mathcal{G}_t}(H) = \frac{\left| \left\{ S_j \in \mathcal{S}^* : d_{\mathcal{G}_t}(S_j) \geq d_{\mathcal{G}_t}(H) \right\} \right|}{|\mathcal{S}^*|} \qquad (14)$$

A low *P* value indicates that the subgraph representing haplotype $H$ is denser compared to other subgraphs of $\mathcal{G}_t$.

The naive way to produce the sample $\mathcal{S}^*$ is to randomly generate subgraphs of $\mathcal{G}_t$ of the size $|H|$. However, SARS-CoV-2 coordinated substitution networks are relatively sparse, and thus many sampled subgraphs will be a priori disconnected and, consequently, also sparse. As a result, such a sampling scheme is inherently biased towards assigning low *P* values to haplotypes corresponding to connected subgraphs and subgraphs with few connected components. Known VOCs and VOIs at most time points have these properties, and thus their statistical significance could be overestimated. This problem can be resolved by sampling only connected subgraphs.

The following numerical example shows why an advanced method for connected subgraph sampling is essential and a naive approach is ineffective. Consider one of the coordinated substitution

networks generated in this study with 1273 vertices and 7329 edges. For a tree (a minimal connected subgraph) with 10 vertices, naive sampling of 1,000,000 samples yielded 188 subgraphs not less dense than the tree, resulting in a $P$ value of 0.000188. In contrast, the $P$ value from connected subgraph sampling is 1, as a tree has the minimal density among all connected subgraphs. Moreover, naive sampling produced only 2 connected subgraphs, making re-normalization with respect to such a small subsample unreliable. Generating a sufficiently large sample of connected subgraphs via naive method is thus impractical due to the enormous naive sample size required.

To sample connected subgraphs, we utilize a more sophisticated randomized enumeration sampling algorithm that follows the network motif sampling scheme introduced in ref. 58. The algorithm assumes that vertices of $\mathcal{G}_t$ are labeled by the unique integers $1, ..., L$, and performs a recursive backtracking. For each vertex $v$ in ascending numerical order, the algorithm iteratively grows a connected subgraph $S$ by adding a randomly chosen new vertex $w$ from the set of allowed extensions $W$. The set $W$ is then updated to include neighbors of $w$ not in the exclusion set $X$. The exclusion set $X$ allows to speed up the calculations and prevent double sampling by excluding (a) neighbors of vertices already in $S$ to avoid multiple additions of the same vertex to $W$ and (b) vertices numbered 1 to $v$ to prevent re-sampling subgraphs that should have been sampled at earlier iterations. The process continues until a subgraph of the desired size $k$ is formed. Sampling for subgraphs containing the vertex $v$ goes on until the predetermined sample count is reached. The full procedure is detailed in Algorithm 1, and the proof of the correctness of this scheme can be found in ref. 58.

If, at some point, the subgraph induced by the haplotype $H$ is disconnected, we replace it with its largest connected component. In this study, for each analyzed coordinated substitution network $\mathcal{G}_t$, the sampling was performed until $M = \min\{3000, \eta_{\mathcal{G}_t}(v)\}$ subgraphs for each vertex $v$ are generated, where $\eta_{\mathcal{G}_t}(v)$ is the total number of connected subgraphs containing $v$. The value of 3000 was selected empirically to provide a sufficient number of sampled subgraphs for all analyzed viral variants.

**Algorithm 1.** Sampling of connected $k$-subgraphs without forbidden pairs

1: **Input:** graph $\mathcal{G}$ with $V(G) = \{1, ..., L\}$, an integer $k$ and the sample size per vertex $M$.
2: **for** $v = 1 : L$ **do**
3: $S \leftarrow \{v\}$; $W \leftarrow N_{\mathcal{G}}(v) \setminus \{1, ..., v\}$; $X \leftarrow N_{\mathcal{G}}(v) \cup \{1, ..., v\}$
4: **global** $M_v = 0$
5: **call** SampleSubgraph($v, S, W, X$)
6: **end for**
 SampleSubgraph($v, S, W, X$)
7: **if** $|S| = k$ **then**
8: **output** $S$, $M_v \leftarrow M_v + 1$ and **return**
9: **end if**
10: **while** $W \neq \emptyset$ and $M_v \leq M$ **do**
11: sample a random vertex $w \in W$ and set $W \leftarrow W \setminus \{w\}$
12: $S' \leftarrow S \cup \{w\}$; $W' \leftarrow W \cup (N_{\mathcal{G}}(v) \setminus X)$; $X' \leftarrow X \cup N_{\mathcal{G}}(w)$
13: **call** SampleSubgraph($v, S', W', X'$)
14: **end while**

### Inference of viral haplotypes

In this subsection, we describe the method for inference of viral haplotypes as dense communities in coordinated substitution networks. Community detection is a well-established field of network science, with numerous algorithmic solutions proposed over the last two decades[59]. Typically (though not always), the collection of communities in a network is defined as a partition[60]. However, in the case of viral genomic variants, there can be overlaps, as observed in known VOCs and VOIs. Additionally, most existing algorithms are heuristics designed to scale to the sizes of extremely large networks rather than to produce optimal

solutions. Viral coordinated substitution networks, although containing hundreds of vertices, are typically smaller than most networks studied in applied network theory. Thus we use our own community detection approach, which extends our previously developed methodology[36]. This approach uses exact algorithms rather than heuristics and is tailored to account for the characteristics of viral data.

Major steps of our computational framework called HELEN (Heralding Emerging Lineages in Epistatic Networks) are depicted in Fig. 6, and the full algorithmic workflow is described by Algorithm 2. For a given time point $t$, HELEN starts by constructing a coordinated substitution network $\mathcal{G}_t$, as described in "Inference of coordinated substitution networks". Then it generates a pool of candidate dense subgraphs of $\mathcal{G}_t$ using Integer Linear Programming (ILP). Finally, it combines generated subgraphs into clusters corresponding to different haplotypes, and infers a haplotype from each cluster.

Generation of dense subgraphs: Our approach is based on a Linear Programming (LP) formulation[61] for finding the densest subgraphs of networks $\mathcal{G}_t$ at each time point $t$. This formulation contains variables $x_i$ for each vertex $i \in V(\mathcal{G}_t)$, variables $y_{ij}$ for each edge $ij \in E(\mathcal{G}_t)$, and the following objective function and constraints:

$$\sum_{ij \in E(\mathcal{G}_t)} y_{ij} \to \max \tag{15}$$

$$y_{ij} \leq x_i, \ y_{ij} \leq x_j, \ ij \in E(\mathcal{G}_t) \tag{16}$$

$$\sum_{i \in V(\mathcal{G}_t)} x_i \leq 1 \tag{17}$$

$$x_i, y_{ij} \geq 0, \ i \in V(\mathcal{G}_t), ij \in E(\mathcal{G}_t) \tag{18}$$

Note that the variables $x_i, y_{ij}$ are continuous rather than integer since it can be shown that the value of the optimal solution of the LP (15)–(18) and the maximum subgraph density of $\mathcal{G}_t$ coincide[61]; furthermore, if $U \subseteq V(\mathcal{G}_t)$ is the vertex set of the densest subgraph, then $(x_i = \frac{1}{|U|}, i \in U; x_i = 0, i \notin U; y_{ij} = \frac{1}{|U|}, i, j \subseteq U; y_{ij} = 0, i, j \nsubseteq U)$ is the optimal solution of (15)–(18). Thus, densest subgraphs of the networks $\mathcal{G}_t$ can be found in a polynomial time.

The single densest subgraph can, however, provide only a single haplotype per time point. We need to produce multiple dense communities to infer multiple haplotypes that could correspond to VOCs and VOIs. So, we generate a pool of candidate dense subgraphs of $\mathcal{G}_t$ as follows. We iterate through a given range of fixed subgraph sizes $k$ from $k_{\max}$ down to $k_{\min}$); at each iteration, we generate a set $\mathcal{S}_k$ of up to $n_{\max}$ densest subgraphs of size $k$ that are not contained in subgraphs generated in the previous iterations. Here $k_{\max}, k_{\min}$ and $n_{\max}$ are parameters of the algorithm. However, finding the densest subgraph of a given size is an NP-hard problem[62,63]. Therefore, for each value of $k$, we use the following Integer Linear Programming formulation:

$$\frac{1}{k} \sum_{ij \in E(\mathcal{G}_t)} y_{ij} \to \max \tag{19}$$

$$y_{ij} \leq x_i, \ y_{ij} \leq x_j, \ ij \in E(\mathcal{G}_t) \tag{20}$$

$$\sum_{i \in V(\mathcal{G}_t)} x_i = k \tag{21}$$

$$\sum_{i \in V(\mathcal{G}_t) \setminus S} x_i \geq 1, \ S \in \bigcup_{k'=k+1}^{k_{\max}} \mathcal{S}_{k'} \tag{22}$$

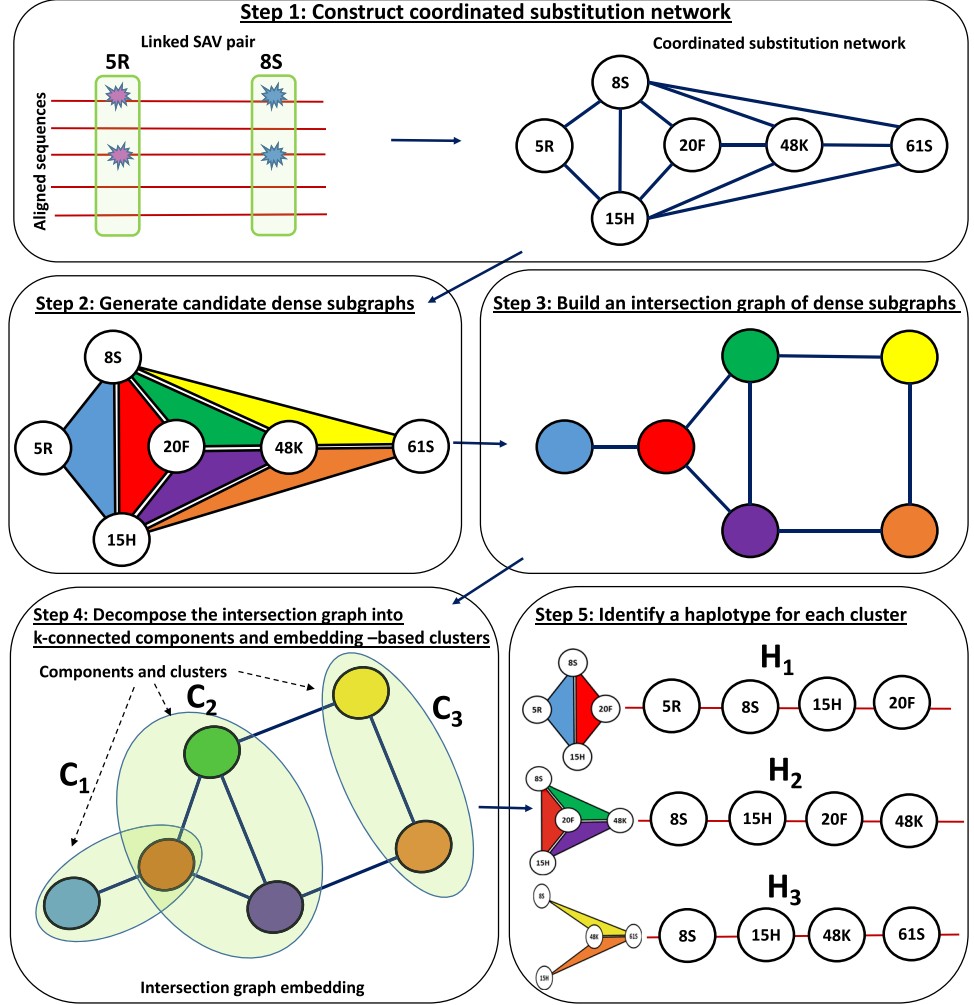

**Fig. 6 | General scheme of HELEN.** Step 1: construction of a coordinated substitution network (CSN) from aligned sequences. Step 2: generation of candidate dense subgraphs of CSN (highlighted in different colors). Step 3: construction of an intersection graph of subgraphs. Each colored vertex represents a subgraph of the same color; two vertices are adjacent whenever the corresponding subgraphs have sufficiently many common vertices (in this example−two). Step 4: decomposition of the intersection graph into clusters (depicted as ovals). Each cluster reflects a single haplotype. Step 5: construction of the haplotype for each cluster. The haplotype is found as a densest community in the union of the CSN subgraphs forming that cluster (e.g., the haplotype $H_1$ is found as the union of the blue and the red subgraphs that form the cluster $C_1$).

$$x_i, y_{ij} \in \{0,1\}, \ i \in V(\mathcal{G}_t), ij \in E(\mathcal{G}_t) \quad (23)$$

Here the constraint (21) sets the size of a dense subgraph, and the constraints (22) ensure that for any subgraph $S$ previously generated, the subgraphs produced in the current iteration must include at least one vertex not in $S$, meaning they should not be subsets of $S$. The problems (15)–(18) and (19)–(23) are solved using Gurobi (Gurobi Optimization, LLC); for the latter, we used an option to continue the search until the pool of up to $n_{max}$ optimal solutions is produced.

Inference of haplotypes from dense subgraphs: Now, let $\hat{S}_t = S_{t,1}, \ldots, S_{t,|\hat{S}_t|}$ be the set of generated densest subgraphs with sizes ranging from $k_{min}$ to $k_{max}$. This set does not necessarily have a one-to-one correspondence with the true haplotypes due to two reasons. First, some haplotypes may consist of more than $k_{max}$ SAVs, so the generated subgraphs only cover parts of these haplotypes. Second, many generated subgraphs overlap significantly, and thus most likely correspond to the same haplotypes.

To obtain full-length haplotypes, we employ the algorithmic pipeline described in detail in steps 3–5 of Algorithm 2 and Fig. 6. Initially, we partition the set of generated dense subgraphs into

clusters such that the subgraphs from each cluster ideally correspond to the same true haplotype. To achieve this, we construct a graph of subgraphs $\mathcal{L}(\hat{S}_t)$, whose edges represent pairs of subgraphs with large overlaps, and split it into clusters using a series of graph clustering techniques. Then, we locate the haplotype for each cluster of subgraphs by finding the densest core community in the union of elements of that cluster.

**Algorithm 2**. HELEN: inference of viral haplotypes using coordinated substitution networks.

**Input**: the set $\mathcal{P}_t$ of aligned viral sequences sampled before or at the time point $t$.

**Output**: the set of haplotypes $\mathcal{H}_t = \{H_{t,1}, \ldots, H_{t,|\mathcal{H}_t|}\}$ designated as potential variants with altered phenotypes.

1) Construct a coordinated substitution network $\mathcal{G}_t$ from sequences $\mathcal{P}_t$, as described in "Inference of coordinated substitution networks".

2) Using the Integer Linear Programming formulation (19)–(23), iteratively generate the set of candidate dense subgraphs $\hat{S}_t = \{S_{t,1}, \ldots, S_{t,|\hat{S}_t|}\}$ of sizes $k \in \{k_{max}, k_{max} - 1, \ldots, k_{min}\}$, so that the elements of $\hat{S}_t$ are not subgraphs of each other.

3) Construct an intersection graph $\mathcal{L}(\hat{\mathcal{S}}_t)$, whose vertex set is $\hat{\mathcal{S}}_t$, and two vertices $S_{t,i}$ and $S_{t,j}$ are adjacent, whenever $|S_{t,i} \cap S_{t,j}| \geq \min\{|S_{t,i}|, |S_{t,j}|\} - 1$. In other words, vertices of this graph of subgraphs are adjacent whenever they have the largest possible intersection.

4) Partition the intersection graph $\mathcal{L}(\hat{\mathcal{S}}_t)$ into clusters $L_{t,1}, \ldots, L_{t,r}$, with each cluster corresponding to a single haplotype. The partition is carried out in stages as follows:

4.1) Split the graph $\mathcal{L}(\hat{\mathcal{S}}_t)$ into connected components and then subdivide each component into $(\kappa + 1)$-connected components, where $\kappa$ denotes the minimum size of a vertex cut (vertex connectivity). To achieve this, we use an algorithm proposed in ref. 64, which computes the vertex connectivity and corresponding vertex cut as the smallest of $(s, t)$-cuts between the fixed vertex $v$ of the minimal degree and its non-neighbors ordered by their distance to $v$, as well as between non-adjacent pairs of neighbors of $v$. The algorithm computes these $(s, t)$-cuts using network flow techniques[65].

We further augmented this algorithm by adding an extra step. Consider a pair of vertices $(s, t)$ for which the minimal vertex cut of size $\kappa_{s,t}$ has been found, and $P_{s,t}^1, \ldots, P_{s,t}^{\kappa_{s,t}}$ are the corresponding internal vertex-disjoint $(s, t)$-paths (which can be found using network flows[65] and whose existence is guaranteed by Menger's theorem[66]). If a vertex $s'$ is adjacent to the internal vertices of all of these paths, then we can exclude the pair $(s', t)$ from further consideration because $\kappa_{s',t} \geq \kappa_{s,t}$. This step significantly accelerates the connectivity calculation for graphs with many high-degree vertices, and the connected components of $\mathcal{L}(\hat{\mathcal{S}}_t)$ typically exhibit this property.

4.2) Suppose that $L_{t,1}, \ldots, L_{t,r'}$ are the components produced at the previous step. Further subdivide each component $L_{t,i}$ as follows: first, find an embedding of the subgraph $\mathcal{L}(\hat{\mathcal{S}}_t)[L_{t,i}]$ formed by the vertices of $L_{t,i}$ into $\mathbb{R}^3$ using a force-directed graph drawing algorithm[67]; second, cluster the obtained embedded graph by a spectral clustering algorithm[68] using the largest Laplacian eigenvalue gap to estimate the number of clusters.

Each cluster produced at steps (4.1)–(4.2) is supposed to contain dense subgraphs corresponding to a single haplotype.

5) For every cluster $L_{t,i}$, we examine the induced subgraph $\mathcal{G}_{t,i} = \mathcal{G}_t[\bigcup_{S_{t,j} \in L_{t,i}} S_{t,j}]$, which consists of the SAVs covered by the dense subgraphs contained in $L_{t,i}$.

5.1) Suppose that $D_{t,i}$ is the sequence of vertex degrees of $\mathcal{G}_{t,i}$. We cluster the elements of $D_{t,i}$ using the $k$-means algorithm and select the subset of vertices $C_{t,i}$ with degrees from the cluster with the largest mean value. The goal of this procedure is to identify the core of $\mathcal{G}_{t,i}$ consisting of high-degree vertices. To choose the number of clusters $k$, we use the gap statistics[69].

5.2) Find the densest subgraph $H_{t,i}$ of $\mathcal{G}_{t,i}[C_i]$ using the LP formulation (15)–(18). If the subgraph is large enough (by default $|H_{t,i}| \geq 5$), then output $H_{t,i}$ as an inferred haplotype.

In addition to the set of haplotypes $\mathcal{H}_t$, Algorithm 2 returns a support $\sigma(H_{t,i})$ for each inferred haplotype, that is defined as a relative number of elements (i.e., candidate dense subgraphs) in the cluster $L_{t,i}$: $\sigma(H_{t,i}) = \sigma_{t,i} = \frac{|L_{t,i}|}{\sum_j |L_{t,j}|}$.

### Reporting summary
Further information on research design is available in the Nature Portfolio Reporting Summary linked to this article.

### Data availability

The findings of this study are based on genomic data available on GISAID; sequence accession numbers can be found at https://epicov.org/epi3/epi_set/230407vq?main=true. The generated secondary data are available at https://github.com/compbel/HELEN(https://doi.org/10.5281/zenodo.10695159).

### Code availability

The code developed and used in this study is available at https://github.com/compbel/HELEN(https://doi.org/10.5281/zenodo.10695159).

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

## Acknowledgements

The authors gratefully acknowledge all data contributors, i.e., the authors and their originating laboratories responsible for obtaining the specimens, and their submitting laboratories for generating the genetic sequence and metadata and sharing via the GISAID Initiative, on which this research is based. The full GISAID acknowledgment list can be found at the tool repository https://github.com/compbel/HELEN (also accessible at https://doi.org/10.55876/gis8.230407vq). P.S. and F.M. were supported by the NSF grants 2047828 and 2212508. A.Z. was supported by NSF grants 2212508, 2316223, 1923679, and NIH grant 1R21CA241044-01A1. S.M. and F.M. were supported by the NSF grants 2041984 and 2316223 and the NIH grant R01AI173172. G.C. was supported by the NSF grant 2125246.

## Author contributions

F.M. developed and implemented algorithms, performed data analysis, wrote the paper. A.Z. developed algorithms, and performed data analysis. S.M. and G.C. performed data analysis. P.S. designed the study, developed and implemented algorithms and models, performed data analysis, wrote the paper, and supervised the project.

## Competing interests

The authors declare no competing interests.
