## [Peer Review File · Nature Communications]

Early Detection of Emerging Viral Variants Through Analysis of Community Structure of Coordinated Substitution NetworksREVIEWER COMMENTS

Reviewer #1 (Remarks to the Author):

This paper deals with analyzing the epistatic relationships driving the evolution of viruses, where SARS-CoV-2 is used as a particular example. The authors present a framework that captures such relationships. Based on this framework, the authors point out how to predict the emergence of new variants. The authors evaluate their claims on data retrieved from GISAID.

The framework presented is truly impressive insofar as it is methodically rich and complex. The framework is further impressive as it indeed gives rise to novel ways to capture and analyze mutations in viruses with a particular view on epistasis. In that respect, I consider the approach quite inspiring.

My major concern about this paper is of formal nature. The authors do not invest much time in delivering notation in a way that allows for full reproducibility. Some notation is just flawed in terms of the basic rules applying for good notation. Some definitions are simply missing. And so on -- I am marking some particularly striking examples of insufficient writing below. However, note that my list is not exhaustive.

MAJOR:

- * The definition of negative and positive epistasis is sloppy and requires precise formalization for being reproducible.
- * What is L in "the space of all possible genotypes (sequence space) forms an L -dimensional hypercube"
- * While one can generally follow, 2.1 lacks formal clarity. What is the definition of "hypercube" you are using here? Fig. 1(c) doesn't really help, because the description is rather sloppy.
- * For example: "Consequently, every sub-hypercube comprises a single local fitness maximum" cannot be understood, because there is no definition of "local fitness maximum". How exactly do you quantify "fitness" here?
- * "let U_0, U_1 and V_0, V_1 be the reference and SAV alleles at two particular genomic positions $U, V \in \{1, \dots, L\}$," -> So, U, V are integers, and indexing integers with 0,1 indicates alleles, i.e. letters of some kind? That is completely confusing, because it

denies canonical rules of notation, despite that it is easy to guess what you want.

* "Let further E_{ij} and O_{ij} be the expected and observed counts of allele pairs (or 2-haplotypes) (U_i, V_j) at a time t ." -> What does time mean here (and in general in your manuscript)? Accurate notation is missing.

* "...have replicative fitnesses f_{ij} ..." -> Definition of fitness still missing, and definition of replicative fitnesses missing in particular

* "transitions between alleles i and j happen at rates q_{Uij} , q_V " -> definitions for transition and rates are missing

* What is the precise definition of a "coordinated substitution network"? (am I overlooking the definition?)

* In summary, please make sure that everyone is able to understand your definitions. These are largely unclear in various places. I haven't mentioned every place where ambiguities occur; please take your time and provide clear definitions and descriptions wherever not yet provided.

* I am missing a clear algorithmic workflow that points out how to predict the emergence of new VOI's / VOC's? Could you please point out how to discover that in terms of a clear algorithmic protocol?

In particular, when following such a protocol, are there potential VOI's / VOC's that never made it into the classification scheme? [to the best of my understanding, administrators of the variant classification scheme did not make use of any mutation interaction framework]

MINOR:

* Citation 53 misses info

* Fig 1 (a): edge colors difficult to see

Reviewer #2 (Remarks to the Author):

Early detection of emerging viral variants is crucial for monitoring pandemics and promoting public health. This paper introduces a framework called HELEN, which aims to detect emerging viral variants. The framework begins by constructing a coordinated substitution network using the observed frequency of single amino acid variants. Two analyses are then performed based on this

network: one focuses on indicating whether a given haplotype is a possible Variant of Concern (VOC) or Variant of Interest (VOI) in the future, while the other aims to infer emerging variants in advance. The results demonstrate HELEN's ability to detect emerging viral variants at an early stage. Overall, the paper is interesting, and the results are comprehensive. However, I have some concerns/questions as below.

The paper starts with an ideal case of having the epistatic network. As this is often unknown, when I read that part, I am a bit confused about what is known and what will be constructed. It will be better to adjust that part to lay out what is known first.

There are several approximations. A very important simplification is to use the coordinated substitution network to represent epistatic networks. Also, instead of cliques, dense subgraphs are used. 2.2 describes the construction of the coordinated substitution networks but it is not clear me what is the "loss" of doing this.

What the impact of the virus mutation rate and the available sequences on the algorithm? For example, if there are many SAVs but limited information about the epistatic network, the derivation of the viable genotypes will be very limited because not all alleles can form cliques. Or the identified viable genotypes only contain at most 1 allele?

SARS-CoV-2 does not mutate as quickly as some other RNA viruses. If there are more mutations, will the network become too large? Can you show the complexity change for a fast-mutating virus?

Theoretically inferring haplotypes is useful. But compared to other methods that mainly focus on mutations at one loci, what new knowledge is derived from this work?

Some more specific questions are listed before. Addressing them will increase the readability and clarify of this work.

Method 2.1:

1. In Fig1. (a), the authors claimed that 'Edges of maximal cliques are displayed in blue, black, and purple.' Please doublecheck the definition of finding maximal clique. Seems to me that only the clique in black in the given example can be called maximal in this figure. Others are just cliques but not the maximum in this figure.

2. While the definitions of negative and positive epistasis are clear, it is a little confusing to understand what negative and positive would look like in the epistatic network. Could you provide further clarification on this matter?

Method 2.2:

3. In section 2.2, you used index 0 and 1 to denote the reference and SAV alleles. However, in Theorem 1, the index of 'U' and 'V' is 2. Could this be a typo? If not, could you please provide a clearer explanation for this indexing discrepancy?

4. In Theorem 1, the inequation of expected counts of allele pairs is always satisfied when $f_{11} = 0$. However, is it possible for the inequation to still be satisfied in some cases when $f_{11} > 0$? If so, it appears that the binomial test may miss some linked SAVs. Could you provide an evaluation of the performance of this binomial test in identifying linked SAVs, if possible?

Method 2.3:

5. Section 2.2 constructs a substitution network for the downstream analysis in sections 2.3 and 2.4.

It is important to note that these two sections are independent, meaning that section 2.3 is not a prerequisite for section 2.4. I would suggest that the authors emphasize the purpose and application scenarios of these two sections, as well as the differences between them, in advance. Personally, I only fully understood the purposes of these sections after reading the results section.

6. I would suggest indicating the meaning of high and low density-based p-values. As I understand it, a low p-value indicates that the subgraph of the haplotype is denser, making it more reliable.

Clarifying this would enhance understanding.

7. In algorithm 1, what is the definition of 'F'? I did not find the definition of 'F' in other parts of the paper. Although I have a rough idea of this algorithm, the details of the extension (e.g., W and X) are a little unclear.

8. In the sentence of the paragraph under Eqn. (10), 'the sampling was performed until $k = \min\{3000, nG_t(v)\}$ ', should 'k' be replaced with '|S*|' or 'M' according to Algorithm 1? It appears that 'k' is representing the subgraph size, but in this context, it denotes the sampling number of subgraphs. This could cause confusion if the symbol has different meanings.

Method 2.4:

9. In algorithm 2, step 2.1 involves splitting the intersection graph into components based on connectivity. It appears that you need to find the minimal vertex cut of size 'connectivity' for the two chosen vertices. Do you need to set a threshold for the size, or can the algorithm determine the size automatically? Additionally, how are the vertices chosen? Many details regarding this part are missing. While it may not be necessary to provide a highly detailed description of the algorithm, I would suggest giving more explanations for each operation in this step, such as the reasons, motivation, and significance behind them.

Result:

1. In Fig. 5, does the gray dashed line denote the time of official designation? The caption does not provide a description of this. It would be helpful to include an explanation of the meaning of the gray dashed line in the caption.

2. Variants may contain a different number of SAVs. Since the analysis is based on the coordinated substitution network, will the number of SAVs in the variants significantly affect the inference of variants from dense communities?

3. Why did HELEN fail to detect 'theta'? Are there any possible reasons for this?

Reviewer #3 (Remarks to the Author):

\section*{Brief Overview}

The authors remark that current viral evolutionary prediction tools concentrate on single amino acid changes (SAVs) or isolated genomic events, and that a haplotype approach, although more challenging might provide better insight. Their claim is that examining community structure in viral coordinated substitution networks can lead to early predictive capabilities when it comes to emergent haplotypes with altered transmissibility. Their framework, HELEN - which they present in the manuscript and validate on SARS-CoV-2 historical data - identifies densely connected communities of alleles and merges them into haplotypes with the ultimate goal of timely variant prediction.

\section*{Detailed overview and section specific comments}

In Section 1\The authors make a compelling case as to why evolutionary forecasting, especially for viral variety, is an important problem. They contextualize it to the SARS-CoV-2 pandemic and remark that although lineage analysis tools are prevalent they are applied, mostly retrospectively.

Moreover, from the perspective of forecasting, although some methods exist, they involve individual mutation considerations, when emerging evidence points to epistatic networks controlling the viral fitness landscape. From here the authors' central thesis follows: that selection acts by haplotypes (collections of correlated mutations) which are, due to combinatorial explosion, difficult to study.

This motivates their framework HELEN, which uses community structure in certain networks to infer haplotypes with altered phenotypes.\bullet\$ \emph{This section is well written, well referenced, and the argumentation of their set-up flows naturally.}

In Section 2.1\The authors present a partial motivation to their approach by way of a negatively constrained epistatic network model for a haploid population. Max-clique decomposition of this network produces a sub-hypercube decomposition of the viable space - by definition - where, within each sub-hypercube only additive or positive epistatic effects occur, yielding a unique local optimum genotype, that can be predicted as the terminus of evolution within its respective sub-cube. The authors use this, as well as practical considerations, of approximating epistatic networks with coordinated substitution ones, to motivate their considerations of dense sub-graphs in the latter.\bullet\$ \emph{This section is a little terse but ultimately understandable. It might benefit from the example being presented in a more rigorous fashion, picking up the binary hyper-cube notational conventions, and providing a more detailed treatment of the clique to sub-cube mapping.}

In Section 2.2\The authors formalize the notion of a coordinated substitution network: a graph over SAVs with edges given by non-random frequency counts. They use a quasispecies model to define this expected count, at discrete times, given random transition rates with fixed replication fitness. They then present Theorem (1), that provides an estimate for non-viable 2-haplotype counts of a specific form. This estimate is used, together with some choices, to construct a criterion for a link to appear in the coordinated substitution network.\bullet\$ \emph{This section may benefit from some reorganisation to improve its flow. A rigorous definition of the coordinated substitution network, with the link criterion devised at the end, could be provided upfront, and then subsequently motivated.}

\bullet\$ \emph{In the context of the quasispecies model they chose, a motivation for why the rates of allelic change are lower than conservation rates could be included. For the concluding remark of Theorem (1), one might mention this only holds for non-trivial conservation rates.}

\bullet\$ \emph{In the paragraph following the proof of Theorem (1), the observed quantities $O_{i,j}$ should be explicitly defined. Finally, some of the key assumptions they make could be further explained. For instance, more details can be provided about why p is a good approximant for 2 -haplotypes under the zero fitness constraint; why we can assume they are binomially distributed; and how the Bonferroni correction enters the picture, when ρ is already free but moderately constrained (as a p-value).}

In Section 2.3\Using the previously devised criterion, the authors now consider temporal coordinated substitution networks of SARS-CoV-2 genomes, and hypothesize that dense vertex induced sub-graphs correspond to VOIs or VOCs. To quantify the statistical significance of this hypothesis, a randomized sub-graph sampling scheme is employed, which accounts for the p-value bias resulting from the sparseness of the SARS-CoV-2 networks.\bullet\$ \emph{This section could be greatly improved. It is difficult to parse at a glance, being time consuming to infer the construction from what has been presented and referenced. It should be rewritten to better explain the workings of Algorithm (1) and how the \mathcal{F} -graph appearing in the formulation of the algorithm comes into the picture. The avoided extension set after one iteration, X , could also be further exemplified, and what the sample size per vertex has to do with the haplotype size must be further elucidated, as it comes into play in the definition

of the proposed p-value.} \bullet\$ \emph{As a note, the replacement of H with the largest connected component when disconnected seems to happen in most cases, as the authors agree that the viral networks considered are already sparse.} \bullet\$ \emph{It is unclear at this point what the "analyzed spike coordinated substitution network" is, and only in section 3.1 do we find that it corresponds to MSAs restricted to the Spike protein region. Also, the choice of stopping criterion \$k\$, being at least \$3000\$ or of the size of the produced sub-graph could be further motivated.} \bullet\$ \emph{Finally, in the beginning, the technical constraints of sparsity that motivate the alternate sampling paradigm could be further elaborated upon. A small explanation could be provided as to why the naive sampling approach cannot simply be amended by re-normalizing.}

In Section 2.4\The authors extend previously developed methodologies for community detection that are well suited to viral data, indexed by time. The classical LP formulation can only be used to optimally compute the densest sub-graph, thus providing a single haplotype per time point, and as VOCs and VOIs have multiplicities as well as overlaps at each point, the authors propose an iterative methodology. The methodology is constrained by densest sub-graph size, and its search is provably NP-hard, hence an ILP formulation is employed and solutions are obtained through methods in the literature. Finally, the authors acquiesce that this methodology potentially generates partial and overlapping haplotypes, and to obtain full length ones a stitching pipeline is further needed. To this end, Algorithm (2) is introduced, where a sequence of (significant) overlap aggregation and subsequent clustering/embedding steps, eventually provide a haplotype set for each time point, and a value, characterizing the support for each haplotype in said set. \bullet\$ \emph{This section is very dense, and might benefit from a further breakdown, into two parts. Wherein, the first part could explain at more length the LP/ILP approach: the setup of the variables, their interpretation, and the optimality results in the constrained case, whilst in the second part the main contribution, Algorithm (2), can be presented in more detail.} \bullet\$ \emph{Figure (2) might benefit from being broken into its sub-figures, and each further annotated and explained during its corresponding section of Algorithm (2).}

In Section 3.1\The authors describe their data preparation procedures. Spike protein region genomes were extracted from GISAID and were sanitized for further processing. Haplotypes for known variants were defined with respect to the standard reference sequence, with some being excluded due to their lengths. As their methodology is influenced by alignment size, only suitably sampled countries were considered, and the test data was further subdivided into tranches based on their reported investigation status. \bullet\$ \emph{This section is well written and self explanatory.}

In Section 3.2\The structure of the S-gene's coordinated substitution networks, that were temporally and spatially annotated, was investigated. It is reported that the majority contain a giant, and converge towards scale-freeness, with Yule-Simon or Pareto degree distributions. This, the authors quote, supports a rich community structure, lending itself to forecasting. \bullet\$ \emph{This section is well written; the figures and the statistical diligence seem to support the authors' claims of a rich structure, in terms of connected cores and sub-graph densities.} \\\newpage

In Section 3.3-.1,.2,.3\The authors posit their central Hypotheses (H1-3): that variants emerge as dense sub-graphs in their construction, that the former correspond to altered phenotype haplotypes, and that the detection thresholds are below significant variant prevalence. This is effected, in terms of precision and recall jointly or at country resolution levels, using a forecasting depth based on WHO designations and peak or cut-off prevalences as well as cumulative frequencies at the earliest detection time. \bullet\$ \emph{The reporting in these sections is factual, and very thorough. Some choices, such as various metrics developed or utilized, some parameter choices, such as 80% f-score thresholds or the 1% prevalence mark could arguably be better motivated, but the sections read very well indeed.}

In Section 3.4\The authors give a description of the compute-hardware used, its performance, and the scalability of their framework. \bullet\$ \emph{This section

is brief but informative. It might benefit from a mention of how the computational performance of the pipe-line compares to other graph community discovery algorithms at the same vertex counts.}In Section 4\The authors present a summary of their work, and concluding remarks to their study. They surmise that there is strong evidence to support their thesis: that dense community structure in coordinated substitution networks is correlated to viral variant dynamics. Their method is contrasted to current variant assignment techniques which scale poorly with sequencing densities. Finally, the framework's shortcomings are disclosed, future augmentation is mentioned, as well as different application areas.}\$\bullet\$ \emph{This section is soberly written and very much to the point.} \section*{Suggested Corrections}

(Page 6) \$\bullet\$ \emph{Strictly speaking, condition (2) should be qualified to $i \neq j$, rather than the indices taking independent values.}(Page 6) \$\bullet\$ \emph{In the formulation of Theorem (1) the 2-haplotype should be (U_1, V_1) not (U_2, V_2) .}(Page 9) \$\bullet\$ \emph{In the formulation of Algorithm (1), The SampleSubgraph recursion is called (in line 5) with parameters out of order.}(Page 10) \$\bullet\$ \emph{omission typo: "Our method produces these communities is as follows".}(Page 10) \$\bullet\$ \emph{index typo: "nmax" should be n_{\max} .}(Page 10) \$\bullet\$ \emph{The bounds on k for the iteration are ambiguously denoted. Suggest $k = k_{\max}$ then $k_{\max}-1, k_{\max}-2, \dots, k_{\min}$.}(Page 11) \$\bullet\$ \emph{In the formulation of constraint (18), S should be a subset not an element of the union.}(Page 11) \$\bullet\$ \emph{index typo: "Split $\mathcal{L}(\hat{S}_t)$ " should be Split $\mathcal{L}(\hat{S}_t)$.}(Page 15) \$\bullet\$ \emph{typo: "emeperical" should be empirical.}

Responses to reviewer comments

December 30, 2023

We sincerely thank the reviewer for highly useful comments and suggestions. All of them were addressed in the revised version of the manuscript and in our replies. The changes in the text are highlighted in **red**.

Please also note that the paper title, the abstract and titles of some sections were shortened to satisfy the journal formatting instructions.

1 Reviewer 1 comments

Comment 1: *The definition of negative and positive epistasis is sloppy and requires precise formalization for being reproducible.*

Reply: More formal definitions of negative epistasis, positive epistasis and fitness landscape were added as follows:

”Consider a population of haploid genotypes $\mathcal{P} = \{g_1, \dots, g_n\}$ with a fixed length L and two potential allelic states 0 and 1 at each locus, where 0 stands for the reference allele and 1 stands for an alternative allele. Each genotype is thus represented as a binary sequence, and all possible 2^L genotypes form a *sequence space* represented as L -dimensional hypercube \mathcal{H} [42], i.e. a graph whose vertices are 0 – 1 sequences of length L , and two vertices are adjacent whenever they differ in a single coordinate.

Each genotype g_i is assigned the *fitness* f_i – a real number that serves as a quantitative measure of its reproductive capacity [43,44]. The function mapping genotypes into the set of their fitness values is referred to as a *fitness landscape* [43].

In our model, the genotypes are subject to *negative epistasis* [45,46]. Following e.g. [44,47], it is defined as the statistical effect where the combined effect of mutations at two specific loci leads to a lower fitness than if these mutations occurred independently, i.e. $f_{11} < f_{10} + f_{01} - f_{00}$, where f_{ij} , $i, j = 0, 1$ are expected fitnesses of genotypes with allelic states (i, j) at the loci. Similarly, positive epistasis occurs when the combined effect of multiple mutations results in a higher than expected fitness, i.e. $f_{11} > f_{10} + f_{01} - f_{00}$ [44,47].”

Comment 2: *What is L in ”the space of all possible genotypes (sequence space) forms an L -dimensional hypercube”*

Reply: The parameter L is defined earlier in the section in the following sentence (slightly modified with respect to the original submission):

"Consider a population of haploid genotypes $\mathcal{P} = \{g_1, \dots, g_n\}$ with a fixed length L and two potential allelic states 0 and 1 at each locus, where 0 stands for the reference allele and 1 stands for an alternative allele"

Comment 3: *While one can generally follow, 2.1 lacks formal clarity. What is the definition of "hypercube" you are using here? Fig. 1(c) doesn't really help, because the description is rather sloppy.*

For example: "Consequently, every sub-hypercube comprises a single local fitness maximum" cannot be understood, because there is no definition of "local fitness maximum". How exactly do you quantify "fitness" here?

Reply: The definitions of a hypercube, fitness, fitness landscape and sequence space were added, please see the reply to Comment 1. The mention of local maximum was removed. The description of the Fig. 1 was extended to include the specific example of a clique of the epistatic network and the corresponding sub-hypercube.

Comment 4: *"let U_0, U_1 and V_0, V_1 be the reference and SAV alleles at two particular genomic positions $U, V \in \{1, \dots, L\}$," So, U, V are integers, and indexing integers with 0,1 indicates alleles, i.e. letters of some kind? That is completely confusing, because it denies canonical rules of notation, despite that it is easy to guess what you want.*

Reply: We agree that indices U_i and V_j are confusing, since it is already mentioned previously that genotypes are represented by binary vectors. Thus, it is more straightforward to denote 2-haplotypes as (i, j) , $i, j = 1, 2$ rather than (U_i, V_j) . The text was modified to incorporate these changes.

Comment 5: *"Let further E_{ij} and O_{ij} be the expected and observed counts of allele pairs (or 2-haplotypes) (U_i, V_j) at a time t ." - \dot{z} What does time mean here (and in general in your manuscript)? Accurate notation is missing.*

Reply: The text was modified as follows:

"Consider a population consisting of N haploid genotypes of length L whose observed abundances change over time points $t = 1, \dots, T$. Let us focus on two particular distinct positions $u, v \in \{1, \dots, L\}$ and consider 4 possible 2-haplotypes $(i, j) \in \{(0, 0), (0, 1), (1, 0), (1, 1)\}$, where 0 and 1 are reference and alternative alleles in u and v respectively. Let also O_{ij}^t be an observed count of 2-haplotypes (i, j) , $i, j = 0, 1$, at a time point $t \in \{1, \dots, T\}$."

Comment 6: *"...have replicative fitnesses f_{ij} ..." Definition of fitness still missing, and definition of replicative fitnesses missing in particular*

Reply: The definition of fitness was added, please see the reply to Comment 1. Adjective "replicative" was removed.

Comment 7: *"transitions between alleles i and j happen at rates q_{ij}^U, q_{ij}^V " definitions for transition and rates are missing*

Reply: The word "transition" here is used a synonym for the word "change" or "mutation", and has the same meaning as, for instance, for Markov chains. The term "rate" was replaced with the term "probability". The text was modified as follows:

"We suppose that viral evolution is driven by mutation and selection, where (a) each 2-haplotype (i, j) has fitness f_{ij} ; (b) each transition (mutation) from the allele k to the allele l at the position u (resp. v) happens with probability q_{kl}^u (resp. q_{kl}^v)."

Comment 8: *What is the precise definition of a "coordinated substitution network"? (am I overlooking the definition?)*

Reply: Coordinated substitution network is defined at the beginning of Subsection 2.2 as follows: "We define a coordinated substitution network as a graph with nodes representing SAVs, and two nodes being adjacent whenever the corresponding non-reference alleles are simultaneously observed more frequently than expected by chance."

Following this comment and the suggestion of the Reviewer 3, this definition is now followed by the formal definition (previously formulated later) as follows:

"Formally, SAVs at positions u and v are adjacent in \mathcal{G}_t (or *linked*), when the following inequality holds:

$$1 - \sum_{i=0}^{O_{11}^t - 1} \binom{N}{i} \left(\frac{O_{10}^t \cdot O_{01}^t}{O_{00}^t \cdot N} \right)^i \left(1 - \frac{O_{10}^t \cdot O_{01}^t}{O_{00}^t \cdot N} \right)^{N-i} \leq \frac{\rho}{\binom{L}{2}}, \quad (1)$$

where ρ is a predefined p -value."

Comment 9: *In summary, please make sure that everyone is able to understand your definitions. These are largely unclear in various places. I haven't mentioned every place where ambiguities occur; please take your time and provide clear definitions and descriptions wherever not yet provided.*

Reply: The Methods section was significantly revised and extended to include definitions and additional explanations. We hope that the introduced modifications make the presentation more clear. In the revised version, we aimed to adopt a level of definitional rigor that is standard in computational biology and population genetics literature intended to be accessible for researchers from both biological and mathematical backgrounds (e.g. [45-47]).

Comment 10: *I am missing a clear algorithmic workflow that points out how to predict the emergence of new VOI's / VOC's? Could you please point out how to discover that in terms of a clear algorithmic protocol?*

Reply: We modified Algorithm 2, so that now it describes the entire algorithmic workflow from the construction of an epistatic network to inference of predicted haplotypes. A schematic depiction of the algorithmic protocol is also presented by Fig. 2.

Comment 11: *In particular, when following such a protocol, are there potential VOI's / VOC's that never made it into the classification scheme? [to the best of*

my understanding, administrators of the variant classification scheme did not make use of any mutation interaction framework]

Reply: The study indeed identified such variants. As outlined in the Results section, HELEN detected a number of haplotypes designated as 'spreading.' These are variants that saw at least a 10-fold increase in prevalence at future time points. These variants did not become VOC/VOI likely due to influence of factors like genetic drift or effective containment via public health interventions preventing them from reaching widespread prevalence. For results on the specificity of HELEN regarding these variants, please refer to pages 22 to 24 and Figure 6F.

Comment 12: *Citation 53 misses info*

Reply: Missing information was added.

Comment 13: *Fig 1 (a): edge colors difficult to see*

Reply: The figure was updated.

2 Reviewer 2 comments

Comment 1: *The paper starts with an ideal case of having the epistatic network. As this is often unknown, when I read that part, I am a bit confused about what is known and what will be constructed. It will be better to adjust that part to lay out what is known first.*

Reply: We added the following introductory paragraph to Methods section. This paragraph provides an overview of its structure and briefly describes what is known and what is to be done in each subsection.

"The major goal of this study is to develop and validate a methodology that, given viral sequences sampled at several time points, infers potentially emerging viral haplotypes by analyzing dense communities of coordinated substitution networks. To achieve it, this section is organized as follows. First, we provide a theoretical justification of the proposed approach by considering an idealized model of an evolving population with given fitness landscape and epistatic network (Subsection 2.1). As epistatic networks are not directly observable, Subsection 2.2 outlines the methodology to infer them from sequencing data. Subsection 2.3 describes our approach to validate the statistical significance of associations between known VOCs/VOIs and dense communities in inferred epistatic networks. Finally, Subsection 2.4 presents the algorithmic framework to de novo infer emerging viral variants."

Comment 2: *There are several approximations. A very important simplification is to use the coordinated substitution network to represent epistatic networks. Also, instead of cliques, dense subgraphs are used. 2.2 describes the construction of the coordinated substitution networks but it is not clear me what is the "loss" of doing this.*

Reply: We acknowledge that coordinated substitution networks, while serving as approximations of epistatic networks, might include elements beyond actual epistatic links, such as those associated with genetic hitchhiking. However, it is important to emphasize that the primary objective of our study is to develop a predictive model, not to delve into the detailed study of epistatic interactions within the SARS-CoV-2 genome. In this context, utilization of coordinated substitution network is unlikely to compromise the accuracy of our methodology, as, for instance, pairs of SAVs connected through genetic hitchhiking could be parts of real emerging variants just as well as pairs linked by epistasis. Furthermore, accurate inference of true epistatic networks is highly challenging, as it requires knowledge of fitness landscapes, which are unobservable and extremely difficult to infer computationally. As a result, relying on coordinated substitution networks is a common approach in the field, as evidenced by studies referenced in our paper.

In general, the links identified by our method are intended as features for our predictive model, rather than as precise individual representations of epistatic interactions. We suggest viewing these links collectively as a statistical ensemble, with our analysis indicating that haplotypes with altered phenotypes tend to have a significantly higher number of potential epistatic pairs compared to background haplotypes. Thus, it is important to note that studies aiming to deeply understand specific SARS-CoV-2 epistatic interactions should integrate more extensive structural data and should constitute separate research projects. We discuss this aspect in the Discussion section of our paper.

As regards to usage of dense subgraphs instead of cliques, we believe that this approach is necessary. This is because epistasis, inherently a statistical property, makes the observation of perfect cliques highly improbable in real biological data. In particular, strict reliance on cliques would make an algorithm non-robust with respect to a noise in data, while consideration of dense subgraphs make estimations more robust. A clique can be easily destroyed by a single missing edge, while a dense subgraph is significantly more stable to edge removal. Therefore dense subgraphs were chosen over cliques in HELEN.

Comment 3: *What the impact of the virus mutation rate and the available sequences on the algorithm? For example, if there are many SAVs but limited information about the epistatic network, the derivation of the viable genotypes will be very limited because not all alleles can form cliques. Or the identified viable genotypes only contain at most 1 allele?*

Reply: We agree that having sufficient number of linked SAVs is essential for sensitivity of the proposed methodology. This, in turn, is affected by the number of sequences, as sufficient coverage of SAV pairs is required to detect statistical links between them. This fact is acknowledged in the paper as follows (please see p. 18 and p. 21):

”Sample size seems to significantly impact the haplotype detection. A positive correlation exists between the number of significantly dense VOCs/VOIs and the number of sequences per country ($\rho = 0.59$, $p = 0.017$). In particular, in the United States, which has the highest number of sequences, all 10 variants

reached significant density”

”Similar to the case with significantly dense subgraphs, sample sizes and geographic diversity influence variant detection. A medium-to-strong positive correlation was observed between the number of sequences per country and the number of variants with positive forecasting depths (Table 1).”

The proposed methodology is not suitable for detection of variants containing a single allele. In this case, we suggest to use one of existing models for detection of spreading mutations cited in our paper. It should be noted, though, that real spreading viral variants are usually defined by multiple alleles. In particular, known Variants of Concern contain from 7 to 30 SAVs.

Comment 4: *SARS-CoV-2 does not mutate as quickly as some other RNA viruses. If there are more mutations, will the network become too large? Can you show the complexity change for a fast-mutating virus?*

Reply: The complexity of most computationally intensive stages of our method depends on the genome length and the number of edges of the epistatic network. In particular, the number of variables in the Integer Linear Programming problem (16)-(20), which constitutes the most time consuming step of our pipeline, is $n+m$, where n and m are numbers of vertices and edges of the epistatic network, respectively. Fast-mutating RNA viruses have relatively short genomes, which makes the proposed methodology scalable. Furthermore, evolutionary studies of such viruses usually concentrate on particular genomic regions, e.g. gag (≈ 500 amino acids), pol (≈ 1000 amino acids) and env (≈ 800 amino acids) for HIV or NS5A (≈ 250 amino acids), NS5B (≈ 125 amino acids) and HVR1 (≈ 30 amino acids) for Hepatitis C virus (HCV) (numbers of loci involved in epistatic interactions should be much lower). Graphs with such numbers of vertices can be processed in reasonable time using modern Integer Linear Programming solvers such as Gurobi. Density of inferred coordinated substitution networks are likely to increase for higher mutation rates, but the degree of increase is hard to ascertain, since there is no datasets of highly mutable viruses with numbers of sequences comparable with that for SARS-CoV-2. We are currently working on a project where HELEN is applied to HCV data, and the networks we obtained there are not dense. This can be a genuine property of epistatic networks, or the consequence of the sample size. Furthermore, there is always a possibility to use lower significance level for the binomial test of SAV linkage, which will make a network more sparse, if needed for efficient calculations.

Comment 5: *Theoretically inferring haplotypes is useful. But compared to other methods that mainly focus on mutations at one loci, what new knowledge is derived from this work?*

Reply: We believe, as we state in the Introduction, that ”given the role of epistasis, it can be argued that selection often acts on combinations of mutations, or haplotypes, rather than on individual mutations. Therefore, effective forecasting should focus on viral haplotypes instead of solely on SAVs.” The emergence of VOCs, like the Omicron variant, which significantly diverged from earlier variants, underscores the non-linearity of viral evolution and suggests that it

cannot be explained by simple gradual accumulation of individual mutations. Therefore, our method, which predicts the impact of combined mutations, offers an advantage over existing approaches that focus on individual mutations. These conventional methods typically rank and prioritize mutations without considering their combined effects, thus limiting their predictive capability.

A practical application of this haplotype-based genomic surveillance is evident in vaccine development. For example, the selection of seasonal influenza vaccines hinges on predicting the dominant lineage for the upcoming season. Usually, these emerging lineages with fitness advantages are characterized by combinations of mutations rather than single mutations. Therefore, the timely identification of Candidate Vaccine Viruses (CVVs) is closely tied to the predictive accuracy of genomic surveillance tools. This principle is equally relevant to SARS-CoV-2 and other pathogens, with the emergence of Omicron variant being the most vivid examples. We added the mention of possible applications of our tool to the end of the Discussion section.

Comment 6: *In Fig1. (a), the authors claimed that 'Edges of maximal cliques are displayed in blue, black, and purple.' Please doublecheck the definition of finding maximal clique. Seems to me that only the clique in black in the given example can be called maximal in this figure. Others are just cliques but not the maximum in this figure.*

Reply: By "maximal clique" we mean "maximal by inclusion", i.e. a clique that is not contained in another clique. Unfortunately, this was not explicitly stated in the text, so we added the full definition to avoid confusion.

Comment 7: *While the definitions of negative and positive epistasis are clear, it is a little confusing to understand what negative and positive would look like in the epistatic network. Could you provide further clarification on this matter?*

Reply: In this particular example, the edges of the network signify the viability of pairs of loci, i.e. the absence of negative epistasis. The edges can correspond to positive epistasis or linear fitness effect. Thus, this theoretical network can be more accurately called a coordinated substitution network, since its definition agrees with the definition of the empirical coordinated substitution network constructed in the following sections. The corresponding adjustment of the text was made.

Comment 8: *In section 2.2, you used index 0 and 1 to denote the reference and SAV alleles. However, in Theorem 1, the index of 'U' and 'V' is 2. Could this be a typo? If not, could you please provide a clearer explanation for this indexing discrepancy?*

Reply: It was a typo, it is now fixed.

Comment 9: *In Theorem 1, the inequation of expected counts of allele pairs is always satisfied when $f_{11} = 0$. However, is it possible for the inequation to still be satisfied in some cases when $f_{11} > 0$? If so, it appears that the binomial test may miss some linked SAVs. Could you provide an evaluation of the performance of this binomial test in identifying linked SAVs, if possible?*

Reply: Theorem 1 establishes a sufficient but not necessary condition, so it is indeed possible that (3) is satisfied when $f_{11} > 0$. Unfortunately, it is hard to directly benchmark the accuracy of the binomial test, since few true pairs of linked SAVs are known. Indirectly, it can be partially validated by comparison with epistatic pairs published in other studies. Specifically, the binomial test applied to the USA dataset recognized 82% of pairs listed in reference [23] and 79% of non-trivial pairs from reference [26] (without considering clusters of consecutive SAVs also reported there). It is not a full validation, since other methods also can be wrong, but significant agreement between them can be an indication of their accuracy. This comparison was added to the paper, please see Subsection 3.2.

It is also important to note that the exhaustive study of all SAV pairs is not the central objective of this study. Therefore, some missed links are unlikely to significantly impact the detection of haplotypes, as the links are analyzed as a statistical ensemble rather than on an individual basis. For a more detailed discussion on this matter, please refer to our response to Comment 2.

Comment 10: *Section 2.2 constructs a substitution network for the downstream analysis in sections 2.3 and 2.4. It is important to note that these two sections are independent, meaning that section 2.3 is not a prerequisite for section 2.4. I would suggest that the authors emphasize the purpose and application scenarios of these two sections, as well as the differences between them, in advance. Personally, I only fully understood the purposes of these sections after reading the results section.*

Reply: We added an introductory paragraph to Results section describing a purpose of each subsection (please see reply to Comment 1). We also added introductory sentences to Subsections 2.3 and 2.4. as follows:

Section 2.3: "We hypothesize that viral haplotypes corresponding to potential VOCs and VOIs form dense subgraphs of coordinated substitution networks. Below we describe the method used to statistically verify this hypothesis."

Section 2.4: "In this subsection, we describe the method for inference of viral haplotypes as dense communities in coordinated substitution networks."

Comment 11: *I would suggest indicating the meaning of high and low density-based p -values. As I understand it, a low p -value indicates that the subgraph of the haplotype is denser, making it more reliable. Clarifying this would enhance understanding.*

Reply: The following sentence with clarification was added:

"A low p -value indicates that the subgraph representing haplotype H is denser compared to other subgraphs of \mathcal{G}_t ."

Comment 12: *In algorithm 1, what is the definition of 'F'? I did not find the definition of 'F' in other parts of the paper. Although I have a rough idea of this algorithm, the details of the extension (e.g., W and X) are a little unclear.*

Reply: \mathcal{F} was an input of the more general version of the algorithm. In the final version of the paper we assume that $\mathcal{F} = \emptyset$, so it is not needed, but was

not removed from Algorithm 1 by mistake. We apologize for the confusion, the error was fixed.

Comment 13: *In the sentence of the paragraph under Eqn. (10), 'the sampling was performed until $k = \min\{3000, \eta_{G_t}(v)\}$ ', should 'k' be replaced with ' S^* ' or 'M' according to Algorithm 1? It appears that 'k' is representing the subgraph size, but in this context, it denotes the sampling number of subgraphs. This could cause confusion if the symbol has different meanings.*

Reply: It is an inaccuracy on our side, k should be replaced by M . It was fixed.

Comment 14: *In algorithm 2, step 2.1 involves splitting the intersection graph into components based on connectivity. It appears that you need to find the minimal vertex cut of size 'connectivity' for the two chosen vertices. Do you need to set a threshold for the size, or can the algorithm determine the size automatically? Additionally, how are the vertices chosen? Many details regarding this part are missing. While it may not be necessary to provide a highly detailed description of the algorithm, I would suggest giving more explanations for each operation in this step, such as the reasons, motivation, and significance behind them.*

Reply: The algorithm is designed to find the vertex cut of the minimum size. Therefore, specifying a threshold is unnecessary, as the algorithm automatically determines the minimal size of the vertex cut (referred to as connectivity). We have updated the algorithm's description to clarify this point.

We also added the description of the pairs of vertices chosen for calculating cuts. Unfortunately, it is hard to provide the brief justification of this choice because it is based on several theorems from the reference [59], as well as adjustments from the reference [58], that together require several pages to formulate. Furthermore, this part of the algorithm was not developed by our team but was adopted from [58]. For a comprehensive justification of this method, we direct readers to the original publication.

Our contribution here is an adjustment aimed at accelerating computations in graphs with many high-degree vertices, and we provide justifications for this adjustment. To delineate our original work from existing research, we modified the text by dividing the description of section 4.1 into two paragraphs.

Comment 15: *In Fig. 5, does the gray dashed line denote the time of official designation? The caption does not provide a description of this. It would be helpful to include an explanation of the meaning of the gray dashed line in the caption.*

Reply: Yes, the dashed lines denote the times of WHO designations. The explanation of this was added.

Comment 16: *Variants may contain a different number of SAVs. Since the analysis is based on the coordinated substitution network, will the number of SAVs in the variants significantly affect the inference of variants from dense communities?*

Reply: HELEN is specifically designed to be able to infer haplotypes with arbitrary numbers of SAVs without need to specify them as input parameters. Furthermore, our results suggest that the number of SAVs do not significantly affect the algorithm’s performance. Indeed, HELEN was able to reconstruct with high accuracy VOCs/VOIs from alpha (7 SAVs) to Omega (30 SAVs). The correlation between VOC/VOI numbers of SAVs and average f -values at detection was not statistically significant ($\rho = 0.17$, $p = 0.65$). This result was added to the paper.

Comment 17: *Why did HELEN fail to detect 'theta'? Are there any possible reasons for this?*

Reply: The following sentence was added to the text:

”Failure to detect Theta variant can be attributed to the fact that 80% of theta cases were observed in Philippines, a country not included in our analysis due to the smaller sample size.”

3 Reviewer 3 comments

Comment 1: *Section 2.1. This section is a little terse but ultimately understandable. It might benefit from the example being presented in a more rigorous fashion, picking up the binary hyper-cube notational conventions, and providing a more detailed treatment of the clique to sub-cube mapping.*

Reply: Section 2.1. and the caption for Fig. 1 were significantly expanded to include more rigorous definitions and an example.

Comment 2: *Section 2.2. This section may benefit from some reorganisation to improve its flow. A rigorous definition of the coordinated substitution network, with the link criterion devised at the end, could be provided upfront, and then subsequently motivated.*

Reply: The section was reorganized, as suggested.

Comment 3: *Section 2.2. In the context of the quasispecies model they chose, a motivation for why the rates of allelic change are lower than conservation rates could be included. For the concluding remark of Theorem (1), one might mention this only holds for non-trivial conservation rates.*

Reply: A motivation for relation between conservation and change rates, was added as follows:

”Mutation probabilities per genomic position per year of most viruses have orders of magnitude between 10^{-3} and 10^{-5} [51]. Thus, we can assume that for time intervals considered in this study the non-negative probability of allelic change is smaller than the of no-change”.

The mention that the rates are non-trivial was also added.

Comment 4: *Section 2.2. In the paragraph following the proof of Theorem (1), the observed quantities $O_{i,j}$ should be explicitly defined.*

Reply: The definition was added.

Comment 5: *Section 2.2. Some of the key assumptions they make could be further explained. For instance, more details can be provided about why p is a good approximant for 2-haplotypes under the zero fitness constraint; why we can assume they are binomially distributed; and how the Bonferroni correction enters the picture, when ρ is already free but moderately constrained (as a p -value).*

Reply: We have clarified in the revised text that p is not employed as an approximation but rather to establish an upper bound for the probability of observing a pair of minor alleles when the associated 2-haplotype is nonviable. A sufficiently low upper bound ensures that the actual probability in question is also low. More detailed derivation of the formula for p was added.

Bonferroni correction is used to account for multiple comparisons between $\binom{L}{2}$ pairs of SAVs being tested for linkage. Given the large number of pairs, some of them can be linked by chance, so we need to account for the possibility of Type 1 error here. The explanation was added.

In modelling the count of genomes with a given pair of minor alleles by a binomial distribution, we followed [36], where a similar approach was successfully used. We added the reference to the text.

To further justify our model choices, it is important to emphasize the applied nature of this study, whose main goal was to develop a framework for detecting emerging viral haplotypes, not to analyze epistatic interactions within the SARS-CoV-2 genome in depth. Thus, the ultimate test of our model and the chosen parameters is how accurately they identify haplotypes. To ensure we minimize the number of false positive linked SAV pairs, which could obscure haplotypes in coordinated substitution networks, we adopted a conservative approach when selecting assumptions and upper bounds. We fully realize that this might result in missing some true epistatic pairs. However, our method aims to use the inferred links as features of our predictive model, not to map every epistatic interaction precisely. Identified links should be treated as a statistical ensemble; our findings suggest that haplotypes with altered phenotypes have significantly more potential epistatic pairs than those in the background. Some missing links are unlikely to affect our algorithm significantly, as evidenced by the validation results presented in the Results section.

Comment 6: *Section 2.3. This section could be greatly improved. It is difficult to parse at a glance, being time consuming to infer the construction from what has been presented and referenced. It should be rewritten to better explain the workings of Algorithm (1) and how the \mathcal{F} -graph appearing in the formulation of the algorithm comes into the picture. The avoided extension set after one iteration, X , could also be further exemplified, and what the sample size per vertex has to do with the haplotype size must be further elucidated, as it comes into play in the definition of the proposed p -value.*

Reply: Section 2.3 was substantially revised in response to this and other comments.

\mathcal{F} was an input of the more general version of the algorithm. In the final version of the paper we assume that $\mathcal{F} = \emptyset$, so it is not needed, but was not removed from Algorithm 1 by mistake. We apologize for the confusion, the error was fixed.

The description of the sampling method was updated to make it more clear and add additional details about the purpose of the avoided extension set X . It also should be noted that, as mentioned in the paper, the sampling scheme implemented in this paper was theoretically introduced in [52]. We describe our implementation of this scheme mainly to allow readers to reproduce our results. Complete formal proof of correctness of the scheme takes 3 pages in [52], and we just refer readers to it.

Sample size is unrelated to the haplotype size, it was selected to be large enough to provide sufficient number of sampled subgraphs ($\approx 3,000,000$) for all known VOC/VOIs analyzed in this study. Increasing it does not significantly affect the results. The explanation was added to the text.

Comment 7: *Section 2.3. It is unclear at this point what the "analyzed spike coordinated substitution network" is, and only in section 3.1 do we find that it corresponds to MSAs restricted to the Spike protein region. Also, the choice of stopping criterion k , being at least 3000 or of the size of the produced sub-graph could be further motivated.*

Reply: The mention of spike is removed from that statement since the described framework is general and can be applied to any genomic data.

The stopping criterion is at least 3000 or the total number of connected subgraphs, not the size of the produced subgraph. This formula is introduced for the sake of mathematical correctness for cases when the total number of connected subgraphs is below 3000. The value 3000 is empirical, please refer to the reply to Comment 6 for its motivation.

Comment 8: *Section 2.3. Finally, in the beginning, the technical constraints of sparsity that motivate the alternate sampling paradigm could be further elaborated upon. A small explanation could be provided as to why the naive sampling approach cannot simply be amended by re-normalizing.*

Reply: If we understand correctly, re-normalization in this context means generating a sample using the naive approach, extraction of subsample of connected subgraphs and re-normalization with respect to this subsample. The following numerical example was added to the paper to demonstrate why connected subgraph sampling is essential and why re-normalization is ineffective.

"Consider one of coordinated substitution networks generated in this study with 1,273 vertices and 7,329 edges. For a tree (a minimal connected subgraph) with 10 vertices, naive sampling of 1,000,000 samples yielded 188 subgraphs not less dense than the tree, resulting in a p -value of 0.000188. In contrast, the p -value from connected subgraph sampling is 1, as a tree has the minimal density among all connected subgraphs. Moreover, naive sampling produced only 2 connected subgraphs, making re-normalization with respect to such a small subsample unreliable. Generating a sufficiently large sample of connected

subgraphs via naive method is thus impractical due to the enormous naive sample size required.”

If we misunderstood the meaning of re-normalization in the comment, we will be happy to provide another explanation.

Comment 9: *Section 2.4. This section is very dense, and might benefit from a further breakdown, into two parts. Wherein, the first part could explain at more length the LP/ILP approach: the setup of the variables, their interpretation, and the optimality results in the constrained case, whilst in the second part the main contribution, Algorithm (2), can be presented in more detail.*

Reply: The section was subdivided, as suggested. It was also augmented and rewritten in response to this and other comments.

Comment 10: *Figure (2) might benefit from being broken into its sub-figures, and each further annotated and explained during its corresponding section of Algorithm (2).*

Reply: We believe that Figure 2 in its current form allows for better understanding of the entire algorithmic workflow, which is essential for readers (please see, for example, Comment 10 by Reviewer 1). Furthermore, such a figure with an overview of an algorithmic workflow is currently standard for bioinformatics papers published in Nature Communications (please see e.g. <https://www.nature.com/articles/s41467-023-44014-3> or <https://www.nature.com/articles/s41467-021-26944-y>).

Comment 11: *Section 3.3. The reporting in these sections is factual, and very thorough. Some choices, such as various metrics developed or utilized, some parameter choices, such as 80% f-score thresholds or the 1% prevalence mark could arguably be better motivated, but the sections read very well indeed.*

Reply: We appreciate the positive feedback. The 80% threshold was selected as the highest round number to allow for a single AA mismatch for the shortest VOC of length 7 AA ($6/7 = 0.85$). The 1% prevalence benchmark was derived from similar studies focusing on the performance of individual mutation prediction tools. Explanations of parameter choices were added to the revised text.

Comment 12: *Section 3.4. This section is brief but informative. It might benefit from a mention of how the computational performance of the pipe-line compares to other graph community discovery algorithms at the same vertex counts.*

Reply: Other community discovery algorithms will certainly be faster, since they are heuristics designed to produce approximate solutions for extremely large networks. In contrast, our method uses an exact ILP-based algorithm, and is designed to produce optimal solutions for real viral coordinated substitution networks that are moderately sized. In genomic surveillance, longer runtimes (measured in hours) are acceptable while the accuracy of the solutions is paramount, particularly as these forecasts can impact public health decisions.

Furthermore, other tools may not be directly comparable with our method due to differing graph density definitions or additional constraints in other methods, such as the necessity for detected communities to be non-overlapping, as seen in references [53,54].

Comment 13: *Suggested Corrections:*

(Page 6) *Strictly speaking, condition (2) should be qualified to $i \neq j$, rather than the indices taking independent values.*

Reply: The error was fixed.

Comment 14: *Suggested Corrections: (Page 6) In the formulation of Theorem (1) the 2-haplotype should be (U_1, V_1) not (U_2, V_2) .*

Reply: The error was fixed.

Comment 15: *Suggested Corrections: (Page 9) In the formulation of Algorithm (1), The SampleSubgraph recursion is called (in line 5) with parameters out of order.*

Reply: The error was fixed.

Comment 16: *Suggested Corrections: (Page 10) omission typo: "Our method produces these communities is as follows".*

Reply: The text was rewritten in response to other comments, so this sentence was removed.

Comment 17: *Suggested Corrections: (Page 10) index typo: "nmax" should be n_{max} .*

Reply: The error was fixed.

Comment 18: *Suggested Corrections: (Page 10) The bounds on k for the iteration are ambiguously denoted. Suggest $k = k_{max}$ then $k_{max} - 1, k_{max} - 2, \dots, k_{min}$.*

Reply: The sentence was rewritten.

Comment 19: *Suggested Corrections: (Page 11) In the formulation of constraint (18), S should be a subset not an element of the union.*

Reply: We believe that the formula presented is accurate. The union in equation (20) (previously equation (18)) represents the set of dense subgraphs generated in earlier iterations for $k' \in \{k + 1, \dots, k_{max}\}$. The specified constraint ensures that for any subgraph S previously generated, the subgraphs produced in the current iteration must include at least one vertex not in S , meaning they should not be subsets of S . To be more clear, we added a description of the ILP constraints to the revised version of the manuscript.

Comment 20: *Suggested Corrections: (Page 11) index typo: "Split $\mathcal{L}(\hat{S}_t)$ " should be Split $\mathcal{L}(\hat{S}_t)$.*

Reply: The typo was fixed.

Comment 21: *Suggested Corrections: (Page 15) typo: "emeperical" should be empirical.*

Reply: The typo was fixed.

REVIEWERS' COMMENTS

Reviewer #1 (Remarks to the Author):

I thank the authors for taking my concerns for very serious: I am very pleased with the applied changes and added illustrations. Everything now appears fully understandable to me.

I stay with my judgment that this is an extremely rich paper, which deserves publication.

Reviewer #2 (Remarks to the Author):

The authors addressed my previous comments. I have no other questions.

Response to reviewers' comments

Reviewer #1 (Remarks to the Author):

I thank the authors for taking my concerns for very serious: I am very pleased with the applied changes and added illustrations. Everything now appears fully understandable to me.

I stay with my judgment that this is an extremely rich paper, which deserves publication.

Reviewer #2 (Remarks to the Author):

The authors addressed my previous comments. I have no other questions.

Response. We express our gratitude to the reviewers for their positive assessment of our paper.